# Diversification of the *Caenorhabditis* heat shock response by Helitron transposable elements

**Jacob M Garrigues, Brian V Tsu, Matthew D Daugherty, Amy E Pasquinelli***

Division of Biology, University of California, San Diego, San Diego, United States

**Abstract** Heat Shock Factor 1 (HSF-1) is a key regulator of the heat shock response (HSR). Upon heat shock, HSF-1 binds well-conserved motifs, called Heat Shock Elements (HSEs), and drives expression of genes important for cellular protection during this stress. Remarkably, we found that substantial numbers of HSEs in multiple *Caenorhabditis* species reside within Helitrons, a type of DNA transposon. Consistent with Helitron-embedded HSEs being functional, upon heat shock they display increased HSF-1 and RNA polymerase II occupancy and up-regulation of nearby genes in *C. elegans*. Interestingly, we found that different genes appear to be incorporated into the HSR by species-specific Helitron insertions in *C. elegans* and *C. briggsae* and by strain-specific insertions among different wild isolates of *C. elegans*. Our studies uncover previously unidentified targets of HSF-1 and show that Helitron insertions are responsible for rewiring and diversifying the *Caenorhabditis* HSR.

**\*For correspondence:**
apasquinelli@ucsd.edu

**Competing interests:** The authors declare that no competing interests exist.

## Introduction

Heat Shock Factor 1 (HSF-1) is a highly conserved transcription factor that serves as a key regulator of the heat shock response (HSR) (*Vihervaara et al., 2018*). In response to elevated temperatures, HSF-1 binds well-conserved motifs, termed heat shock elements (HSEs), and drives the transcription of genes important for mitigating the proteotoxic effects of heat stress. For example, HSF-1 promotes the expression of heat-shock proteins (HSPs) that act as chaperones to prevent HS-induced misfolding and aggregation of proteins (*Vihervaara et al., 2018*). More recently, HSF-1 has been identified as having non-canonical roles in transcriptional programs distinct from the HSR, such as during development (*Li et al., 2016*) and carcinogenesis (*Mendillo et al., 2012*), underscoring the importance of this ancient transcription factor.

Due to the central role HSF-1 has in launching the HSR, much work has been done to determine direct HSF-1 targets by identifying associated genes under different conditions (*Gonsalves et al., 2011*; *Guertin and Lis, 2010*; *Li et al., 2016*; *Mahat et al., 2016*; *Mendillo et al., 2012*; *Solís et al., 2016*). However, most of these studies have focused on HSF-1 binding sites that do not overlap with repetitive sequences such as transposons, potentially leaving a large gap in our understanding of the types of genes that are part of the HSR. Indeed, in previous work, we serendipitously discovered that HSF-1 promotes the expression of a full-length autonomous transposable element (TE) known as a Helitron during heat shock (*Schreiner et al., 2019*). Helitrons are a type of DNA transposon that are unique due to their replication and subsequent transposition occurring via a rolling-circle 'copy-and-paste' mechanism, which includes replication in place that results in a circular double-stranded DNA intermediate (*Grabundzija et al., 2018*; *Kapitonov and Jurka, 2007*). While Helitrons were initially discovered in plants and *C. elegans* (*Kapitonov and Jurka, 2001*), they have since been identified in a diverse range of species, including humans (*Kojima, 2018*; *Thomas and Pritham, 2015*).

Transposable elements (TEs) are well-known agents of evolution that have the ability to rearrange genomes by moving DNA sequences from one region to another (*Feschotte and Pritham, 2007*). By providing regulatory sequences to regions within or near genes, these rearrangements can affect gene expression in *cis* (*Bourque et al., 2018*; *Chuong et al., 2017*). As a result, new genes can rapidly be incorporated into existing regulatory networks in a single event, bypassing a potentially lengthy process requiring multiple events to occur in both *cis* and *trans* (*Sundaram and Wang, 2018*). In the human genome, TEs have been proposed to supply a substantial number of functional transcription factor binding sites (*Jiang and Upton, 2019*; *Jordan et al., 2003*; *Sundaram et al., 2014*), underscoring their potential for shaping transcriptional programs. In a particularly striking example, endogenous retrovirus (ERV) TEs were found to harbor binding sites for core transcription factors in the mammalian interferon response and, in some cases, provide interferon-inducible enhancers for nearby genes (*Chuong et al., 2016*). Furthermore, systematic analyses of a set of primate-specific TEs revealed widespread enhancer roles in human embryonic stem cells (*Fuentes et al., 2018*; *Pontis et al., 2019*); a similar functional study of mouse TEs identified a small fraction that seem to act as enhancers in early development (*Todd et al., 2019*).

As TEs have a propensity to mobilize upon cellular stress (*Chuong et al., 2017*), this plasticity provides a potential means to increase genomic diversity and facilitate adaptation to harsh environments (*McClintock, 1984*). For example, in fission yeast, various stress conditions increase the mobilization of the Ty3/gypsy family retrotransposon *Tf1*, and new insertions affect the expression of adjacent genes that promote stress resistance (*Esnault et al., 2019*). Similarly, in *Arabidopsis*, the *copia*-type retrotransposon *ONSEN* becomes transcriptionally active during thermal stress and bestows heat-shock-responsiveness to genes near new insertions (*Cavrak et al., 2014*; *Ito et al., 2011*). Thus, TEs play diverse and widespread roles in shaping stress responses at the transcriptional level.

Our finding that a Helitron TE includes HSEs bound by HSF-1 during heat shock (HS) prompted us to map previously published HSF-1 binding data to repetitive regions in the *C. elegans* genome. In doing so, we aimed to determine the extent of HSEs associated with Helitron elements and to obtain a more complete picture of HSF-1-mediated gene induction during heat shock in *C. elegans*. We found that over half of the HSEs in the *C. elegans* genome reside within Helitrons. Consistent with the Helitron-embedded HSEs being functional, many acquire HSF-1 and display increased RNA polymerase II (Pol II) occupancy after HS and are located near up-regulated genes. Unlike evolutionarily conserved and canonical HSR target genes (controlled by Helitron-independent HSEs or Hin-HSEs), we found that Helitron-acquired HSEs (Hac-HSEs) control expression of specific G protein-coupled receptor (GPCR) and collagen genes, revealing numerous previously unidentified HSF-1 targets in *C. elegans*. Moreover, we detected Hac-HSEs in all *Caenorhabditis* genomes analyzed. Interestingly, we found that different genes appear to be recruited into the HSR by species-specific Helitron insertions in *C. elegans* versus *C. briggsae*. Furthermore, we found variability in Helitron position and, therefore, HS inducibility of individual genes, even among wild isolates of *C. elegans*. These findings suggest that recent mobility of Helitrons has altered the HSF-1-driven transcriptional response within the *C. elegans* population. Our studies reveal that Helitrons have incorporated new genes into the HSR in numerous *Caenorhabditis* species and provide a striking example of the ability of transposable elements to act as agents of evolution by rewiring transcriptional responses and increasing variation within and among populations.

## Results

### HSF-1 binds HSEs located within Helitrons

Our previous work showed that full-length copies of the DNA transposon *Helitron1_CE* contain HSF-1 binding motifs (termed heat shock elements, or HSEs) and are bound by HSF-1 during heat shock (*Schreiner et al., 2019*). Additionally, mRNA encoding the *Helitron1_CE*-embedded Rep-Helicase transposase was shown to be up-regulated during heat shock in an HSF-1-dependent manner (*Schreiner et al., 2019*), suggesting that some HSEs within Helitrons are functional and serve to promote expression of mRNAs. We therefore asked whether other types of Helitrons in the *C. elegans* genome associate with HSF-1 and contain HSEs. There are relatively few full-length copies of Helitrons in the genome and even fewer autonomous versions that are predicted to encode for a

transposase (*Hubley et al., 2016*; *Kapitonov and Jurka, 2001*). Instead, the majority of the regions identified as Helitrons in the genome are non-autonomous and fragments of full-length versions, representing insertion events that have subsequently decayed throughout evolution. To determine if other types of Helitrons (both full-length and partial copies) associate with HSF-1 under non-heat shock (NHS) and heat shock (HS) conditions, publicly available HSF-1 ChIP-seq data (*Li et al., 2016*) were mapped to the *C. elegans* genome allowing for mapping to, but preventing pileup at, repetitive regions (see Materials and methods for details). After calling peaks and determining their summits, these regions were compared to annotated Helitrons in a genome-wide manner (UCSC Genome Browser RepeatMasker track, assembly ce11) to identify HSF-1-bound Helitrons. Similar to our observation with *Helitron1_CE*, four other types of Helitrons display HSF-1 binding in NHS conditions and increased binding during HS conditions: *HelitronY4_CE*, *HelitronY1_CE*, *HelitronY1A_CE*, and *Helitron2_CE* (*Figure 1A*).

To determine the sequence within Helitrons bound by HSF-1, MEME (*Bailey and Elkan, 1994*) was used to extract *de novo* motifs from the Helitron-overlapping HSF-1 peak summits observed during HS conditions (*Figure 1B*). For comparison, HSEs were also extracted from HSF-1 summit regions observed during HS that do not overlap with any repetitive sequences (*Figure 1B*). As expected, HSF-1 peak summits within Helitrons were enriched for HSE-like motifs that were determined to be significantly similar to the repeat-masked HSE motif using the motif-comparison tool Tomtom (*Gupta et al., 2007*) (q-value <1.0e-03). Interestingly, the extended sequences in these HSE-like motifs are consistent with tandem HSEs overlapping with one another, suggesting they are repetitive in nature. Additionally, a comprehensive search for the presence of 130 experimentally determined *C. elegans* transcription factor binding motifs (*Matys et al., 2006*; *Narasimhan et al., 2015*; *Weirauch et al., 2014*) in Helitrons showed that the most enriched was the HSE (*Supplementary file 1*). Taken together, these findings reveal that Helitrons are enriched for HSEs and associate with HSF-1.

## Helitrons harbor the majority of HSEs found in the *C. elegans* genome

To examine the occurrence of HSEs within different classes of Helitrons, the positions of HSEs throughout the genome were identified using FIMO (*Grant et al., 2011*) (q-value <0.05) with the MEME-derived output from repeat-masked HSF-1 summits. Many, but not all, individual Helitrons contain HSEs, ranging from 2% of the *Helitron2_CE* class to 44% of the *HelitronY1_CE* class, yielding a total of 953 HSE-containing Helitrons (*Figure 1C*). While most HSE-containing Helitrons harbor between one and ten HSEs, eight copies of *HelitronY4_CE* possess more than 100 HSEs (*Figure 1D*), highlighting the potential of Helitrons to provide significant numbers of HSEs to the genome. Indeed, more than 10,000 total HSEs are found within these five Helitron classes (*Figure 1D*).

To determine whether Helitrons are significantly enriched for HSEs relative to the rest of the genome, the proportion of the genome annotated as Helitron was compared to the proportion of genomic HSEs found within Helitrons. As annotated Helitron positions often overlap with one another, summing these HSE numbers would inflate the actual number of Helitron-residing HSEs. After condensing any Helitrons with overlapping positions into single regions, 1.7% of the *C. elegans* genome (1.68/100.27 Mb) was determined to be Helitron (*Figure 1E*), which is consistent with the original estimate of 2% (*Kapitonov and Jurka, 2001*). Strikingly, we found that 61.4% (11,527/ 18,766) of genomic HSEs reside within Helitrons, while only 38.6% are found outside of Helitrons, resulting in a significant enrichment (p<1.0e-15, binomial test) (*Figure 1F*). These findings show that Helitrons provide the majority of HSEs to the *C. elegans* genome and, hereafter, we refer to these as Helitron-acquired HSEs (Hac-HSEs). The remaining HSEs are referred to as Helitron-independent HSEs (Hin-HSEs).

The observation that almost two-thirds of genome-wide HSEs reside within 1.7% of genomic space suggested that Hac-HSEs exist within high-density clusters. Indeed, while Hin-HSEs are evenly distributed within 1 kb bins along the length of chromosome II (chrII), high numbers of Hac-HSEs are clustered at distinct loci (*Figure 1G*); this pattern is consistent across all six *C. elegans* chromosomes (*Figure 1—figure supplement 1*). In summary, we find that Helitrons are enriched for HSEs that are often clustered (p<1.0e-15, Fisher's exact test), and high HSE densities seem to be restricted to remnants of prior Helitron insertions.

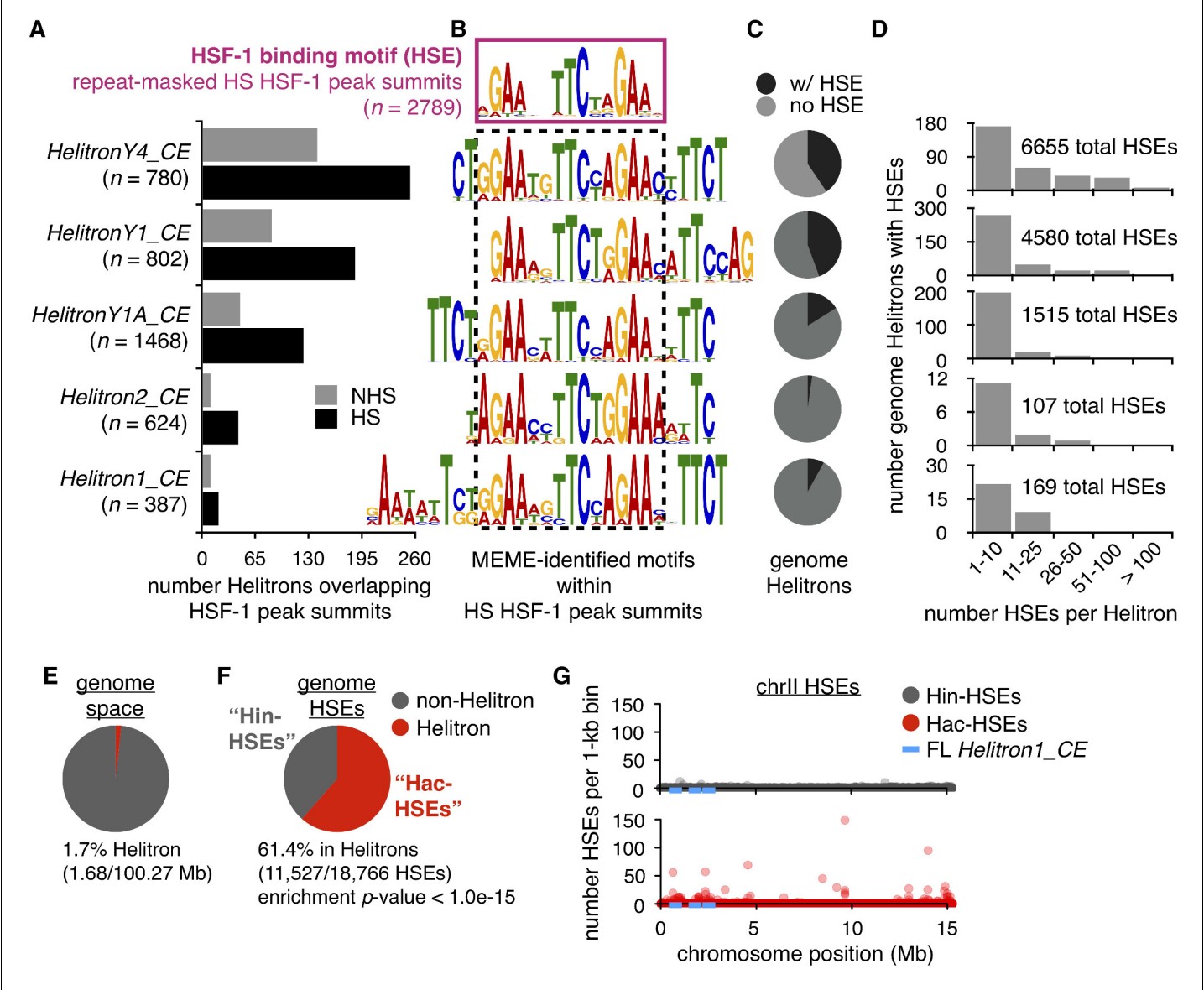

**Figure 1.** Helitrons contribute a significant number of HSEs to the *C. elegans* genome. (**A**) Number of Helitrons that overlap with 101 bp regions centered around HSF-1 peak summits (ChIP-seq data from *Li et al. (2016)*; peaks called using MACS2, *Zhang et al., 2008*) in non-heat shock (NHS) and heat shock (HS) conditions. Helitron coordinates were obtained from the RepeatMasker track downloaded from the UCSC Genome Browser (genome assembly ce11, *Kent et al., 2002*). (**B**) *De novo* motifs extracted from 101 bp regions centered around HS HSF-1 peak summits using MEME (*Bailey and Elkan, 1994*). The HSE motif extracted from all repeat-masked HS HSF-1 peak summits is boxed with a solid purple line, while HSE-like motifs found within non-repeat-masked summits that overlap with Helitrons are boxed with a dashed black line. For repeat-masked HSF-1 peak summits, the displayed HSE motif is the most significant motif found. For Helitron-overlapping summits, displayed HSE-like motifs are within the top 10 significant motifs identified and are significantly similar to the HSE motif extracted from repeat-masked summits using the Tomtom motif comparison tool (q-value <1.0e-03) (*Gupta et al., 2007*). (**C**) Pie chart displaying the proportions of all genomic Helitrons in which HSEs are present (black) or absent (gray). HSEs were identified by scanning the genome with FIMO (q-value <0.05) (*Grant et al., 2011*) using MEME-derived output from repeat-masked HSF-1 peak summits. (**D**) Histograms displaying binned numbers of HSEs within all HSE-containing Helitrons found in the genome. Total numbers of HSEs that reside within each Helitron type are also shown. (**E**) Pie chart displaying the proportion of the *C. elegans* genome that is annotated as Helitron (red) or not (gray). (**F**) Pie chart displaying the number of HSEs in the *C. elegans* genome that reside within (red, hereafter referred to as Hac-HSEs) or outside of (gray, hereafter referred to as Hin-HSEs) annotated Helitrons. HSEs are significantly enriched within Helitron sequences (p<1.0e-15, binomial test). (**G**) Numbers of Hin- (gray) and Hac-HSEs (red) within 1 kb bins along the length of chromosome II (chrII). The approximate positions of full-length (FL) *Helitron1_CE* elements are shown using blue lines.

The online version of this article includes the following figure supplement(s) for figure 1:

**Figure supplement 1.** Genome-wide distributions of Hin- and Hac-HSEs.

## HSF-1 bound Helitron HSEs are enriched near genes up-regulated during heat shock

We next asked whether Hac-HSEs provide nearby genes with the ability to be induced upon heat shock. First, we identified HSF-1 binding events that overlapped Hac-HSEs or Hin-HSEs, using publicly available ChIP-seq data (*Li et al., 2016*). Similar numbers of HSF-1 peak summits during HS conditions were observed to contain Hac-HSEs ($n$ = 556) and Hin-HSEs ($n$ = 578) (*Figure 2A*). For both of these types of HSEs, the levels of HSF-1, as well as RNA polymerase II (Pol II), were generally higher under HS compared to non-HS conditions (*Figure 2A*). These results show that, like canonical HSEs, Helitron-embedded HSEs have the potential to promote transcription in response to HS via recruitment of HSF-1 and RNA Pol II.

We then examined whether protein-coding genes near HSF-1-bound Hac-HSEs are up-regulated during heat shock. Transcriptional profiling (RNA-seq) was performed in NHS- and HS-treated L2 stage worms (matched to conditions used for published HSF-1 and Pol II ChIP-seq data) (*Li et al., 2016*), and the positions of up-regulated genes were compared to HSF-1-bound Hac-HSEs as well as Hin-HSEs (*Figure 2B*). After 30 min of HS at 34°C, 1462 protein-coding genes were significantly up-regulated by at least two-fold compared to NHS conditions at 20°C (*Supplementary file 2*). From the list of up-regulated genes, 119 were found within 2.5 kb of sequence downstream of an HSF-1-bound HSE. Of these, 60 genes were found within 2.5 kb of sequence downstream of a Hac-HSE bound by HSF-1, resulting in a significant enrichment over the expected background ($p<1.0e-15$, $\chi^2$ test) (*Figure 2B*). In addition to one gene containing HSF-1 bound Hac- and Hin-HSEs, the remaining 59 genes were found proximal to Hin-HSEs, resulting in a total of 60 genes that showed a similar enrichment over the background expectation ($p=1.0e-09$, $\chi^2$ test) (*Figure 2B*). Taken together, these data indicate that Helitrons provide functional HSF-1 binding sites to nearby genes, poising them to respond to heat shock.

## Helitron-acquired and -independent HSEs have distinct features

We next asked if different features were associated with heat shock responsive genes regulated by Hac- and Hin-HSEs. First, we compared a representative Hac-HSE gene, *col-88*, to a canonical heat-shock induced gene, *hsp-70*, which lacks a Helitron in its promoter region. Consistent with the overall trend of a high abundance of HSEs in Helitrons (*Figure 1F–G*), the *HelitronY1_CE* element upstream of *col-88* contains 10 HSEs (*Figure 2C*), whereas *hsp-70* only has two HSEs in its promoter (*Figure 2D*). We also observed that, despite the prevalence of HSEs proximal to *col-88*, HSF-1 occupancy before and after heat shock is lower than that observed for *hsp-70* (*Figure 2C–D*). To determine if these differences were consistently observed between Hac-HSE and Hin-HSE HSR genes, we compared their numbers of proximal HSEs and expression levels before and after heat shock (*Supplementary files 3* and *4*). Reflecting the differences observed for *col-88* and *hsp-70*, we found that Hac-HSE genes in general have significantly more HSEs within 2.5 kb upstream of their transcriptional start sites compared to Hin-HSE genes (*Figure 2E*). While the overall fold-induction of Hac-HSE and Hin-HSE genes in response to HS is similar, mRNA expression levels are generally lower for Hac-HSE versus Hin-HSE genes during NHS and HS conditions (*Figure 2F*). Consistent with this observation, Hac-HSE genes exhibit less Pol II coverage relative to Hin-HSE genes both before and during HS (*Figure 2G* and *Supplementary files 3* and *4*). To determine whether this lower overall level of expression of Hac-HSE genes might be due to repressive chromatin environments, we used publicly available data (*Ho et al., 2014*) to measure histone modifications associated with transcriptional repression (H3K27me3 and H3K9me3) and activation (H3K4me3 and H3K36me3) (*Ahringer and Gasser, 2018*). Indeed, Hac-HSE genes have significantly higher levels of the repressive and lower levels of the active histone marks when compared to Hin-HSE genes (*Figure 2H* and *Figure 2—figure supplement 1*) and *Supplementary files 3* and *4*). This signature of histone modifications at Helitron sequences may be due to small-RNA-mediated genome-surveillance mechanisms that transcriptionally repress repeat sequences in *C. elegans* (*Ashe et al., 2012*; *Lee et al., 2012*; *Mao et al., 2015*; *Shirayama et al., 2012*). These differences between Hac-HSEs and Hin-HSEs in terms of number of proximal HSEs, expression levels during both NHS and HS conditions, and presence in repressive chromatin environments, support the notion that Helitron-associated genes represent a distinct class of HSR genes from the classical Helitron-independent HSR genes.

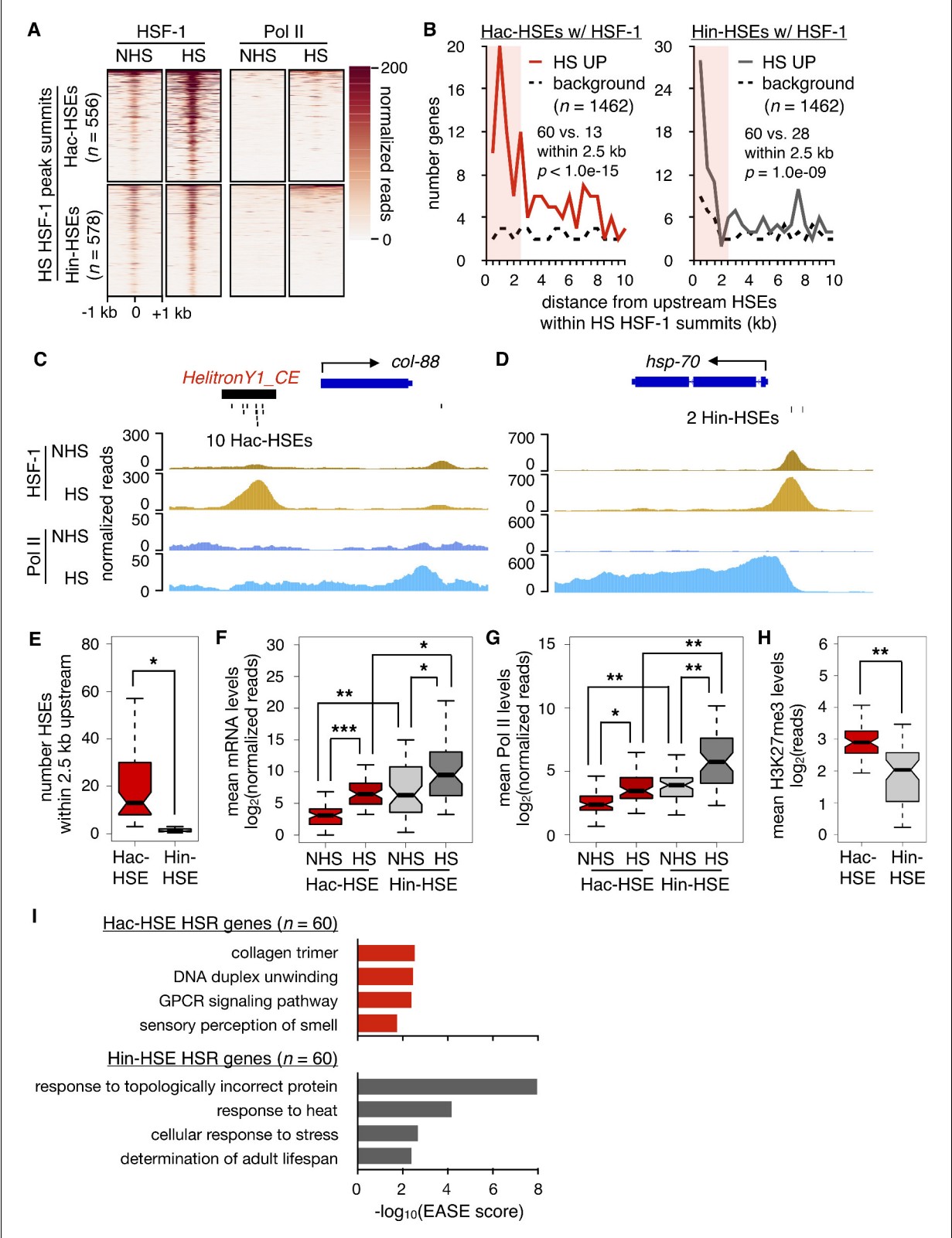

**Figure 2.** HSR genes with adjacent Hac- or Hin-HSEs display different properties. (**A**) Heatmap displaying normalized (see Materials and methods for details) HSF-1 and Pol II ChIP-seq reads over 2 kb regions centered on Hac-HSE- and Hin-HSE-containing HS HSF-1 peak summits. ChIP data are from *Li et al. (2016)*. (**B**) Number of genes within 500 bp bins versus distance from upstream Hac- and Hin-HSEs that reside within HS HSF-1 peak summits. HS UP genes (*n* = 1462) are those with a fold change after HS >2 and an adjusted p-value<0.01 determined using DESeq2 (*Love et al., 2014*). For Hac-
*Figure 2 continued on next page*

*Figure 2 continued*

HSEs, 60 HS UP (red solid line) vs. 13 background (black dashed line) genes reside within 2.5 kb, resulting in a significant enrichment (p<1.0e-15, $\chi^2$ test). For Hin-HSEs, 60 HS UP (gray solid line) vs. 28 background (black dashed line) genes reside within 2.5 kb, resulting in a significant enrichment (p=1.0e-09, $\chi^2$ test). HS UP genes that have HSEs within HSF-1 peak summits less than 2.5 kb upstream from annotated transcriptional start sites are considered to be part of the heat-shock response (HSR genes). (C) Genome browser screenshot of normalized HSF-1 (goldenrod) and Pol II (blue) ChIP-seq reads (note scale differences) in the vicinity of the Hac-HSE HSR gene *col-88*. The 10 Hac-HSEs located within the *HelitronY1_CE* sequence found in the upstream promoter region are shown. (D) Genome browser screenshot of normalized HSF-1 and Pol II ChIP-seq reads (note scale differences) in the vicinity of the Hin-HSE HSR gene *hsp-70*. The two Hin-HSEs located in the upstream promoter region are shown below the gene structure. (E) Notched boxplot showing distributions of the number of HSEs located within 2.5 kb upstream of Hac- (red, *n* = 60) or Hin-HSE (gray, *n* = 60) HSR genes. Notches represent 95% confidence intervals of the median. (F) Notched boxplot showing distributions of DESeq2-normalized steady-state mRNA levels during NHS and HS conditions for Hac- or Hin-HSE HSR genes. As some Hac-HSE HSR genes were undetectable under NHS conditions, a single read was added to all genes to allow for log transformation. (G) Notched boxplot showing distributions of normalized Pol II ChIP-seq reads obtained during NHS and HS conditions over Hac- or Hin-HSE HSR gene bodies. (H) Notched boxplot showing distributions of H3K27me3 ChIP-seq reads using publicly available data obtained from L3 stage worms during NHS conditions (*Ho et al., 2014*) over Hac- or Hin-HSE HSR gene bodies. H3K27me3 data were obtained from the modMine database (intermine.modencode.org) (*Contrino et al., 2012*) under accession number modEncode_5051. For panels E-H, a single asterisk represents p<1.0e-04, double asterisks represent p<1.0e-08, and triple asterisks represent p<1.0e-12 (Welch Two Sample *t*-test). (I) Significantly enriched gene ontology (GO) terms identified using DAVID (*Huang et al., 2009a*; *Huang et al., 2009b*) for Hac-HSE (red) and Hin-HSE (gray) HSR genes.

The online version of this article includes the following figure supplement(s) for figure 2:

**Figure supplement 1.** HSR genes with adjacent Hac- or Hin-HSEs display different levels of histone modifications associated with gene expression or repression.

Previous studies have shown that canonical HSF-1 protein-coding gene targets have roles in mediating the effects of heat stress, as well as promoting longevity (*Li et al., 2017*; *Li et al., 2016*). Consistent with these studies, Hin-HSE genes show an enrichment of gene ontology (GO) terms related to canonical HSR targets, such as response to topologically incorrect protein, response to heat, and determination of adult lifespan (*Figure 2I*). In contrast, Hac-HSE genes are associated with distinct GO enrichment terms, namely collagen trimer, G protein-coupled receptor (GPCR) signaling, and sensory perception of smell (*Figure 2I*), possibly reflecting more recent integration into this stress response. Enrichment for the GO term 'DNA duplex unwinding' derives from the five full-length autonomous Helitrons identified as Hac-HSE genes and reflects the helicase function of Helitron transposases (*Figure 2I*). Differing enrichment of GO terms further demonstrates that Hac- and Hin-HSE genes comprise two distinct classes of HSR genes. While Hac- and Hin-HSE genes have distinct GO enrichment terms, several genes in both categories have been previously implicated in stress response pathways based on RNAi and mutant phenotypes (*Supplementary files 3* and *4*).

We also asked if Hac- and Hin-HSE genes might show different signatures of selection by computing Tajima's D, which is used to determine genetic variation within a population (*Tajima, 1989*). Using publicly available *C. elegans* wild isolate variation data available from the *C. elegans* Natural Diversity Resource (CeNDR) (*Cook et al., 2017*), we determined Tajima's D scores over HSF-1 peak summits that overlap HSEs upstream of Hac- and Hin-HSE genes induced by HS (*Supplementary files 3* and *4*). This analysis did not yield a striking overall difference in the average scores for Hac- versus Hin-HSE genes. It did, however, reveal seven Hac-HSE genes (*col-162, str-30, ZK1053.1, ZK1053.2, srt-28, R02C2.6, oac-55*) and three Hin-HSE genes (*F57A8.1, Y38H6C.8, bath-36*) that have scores below the 5% quantile of genome-wide Tajima's D scores (−2.27), raising the possibility of selection.

## Helitrons provide substantial numbers of HSEs to other *Caenorhabditis* genomes

Our discovery of a large number of Hac-HSEs in *C. elegans* raised the possibility that other species have Helitron-associated HSEs in their genomes. To investigate this, all genomes with annotated Helitrons available from the UCSC Genome Browser (*Casper et al., 2018*; *Kent et al., 2002*) were searched for HSEs, and the numbers and proportions of HSEs found within and outside of Helitrons were determined (*Supplementary file 5*). To prevent possible species-specific biases found in *C. elegans* HSEs from influencing our results, we scanned for HSEs using the canonical species-independent sequence NGAANNTCCNNGGAN and its reverse complement (*Perisic et al., 1989*). As a

result, the total number of HSEs identified in *C. elegans* and the proportion that reside within Helitrons differs in this analysis from what is reported in *Figure 1* (*Figure 3A*, *Supplementary files 5* and *6*). While a variety of species across different phyla have sequences annotated as Helitron, enrichment of HSEs within Helitrons is limited to only the three *Caenorhabditis* species represented in this dataset (*C. elegans, C. briggsae* and *C. brenneri*) (*Figure 3A*). These data indicate that the association of Helitrons with large numbers of HSEs is not a universal phenomenon and may be restricted to *Caenorhabditis* species.

In *C. elegans*, we observed that high-density clusters of HSEs exclusively reside within Helitrons (*Figure 1G*). To analyze the frequency of HSEs within the genomes of other species and to test whether high-density clusters of HSEs may reside outside of Helitrons in other species, we calculated the number of HSEs found within 1 kb bins in all species with annotated Helitrons available from the UCSC Genome Browser (*Supplementary file 5*). When plotting the number of bins with HSEs versus the number of HSEs per bin, curves associated with the three *Caenorhabditis* genomes stand out from all other examined species (*Figure 3B*). This finding suggests that the clustering of HSEs is a signature of Hac-HSEs and that high-density HSE clusters do not exist outside of Helitrons.

To determine the extent to which this Helitron-HSE association exists within the *Caenorhabditis* genus, we identified HSEs in 27 publicly available *Caenorhabditis* genomes for which Helitrons are annotated and determined the total numbers and proportions of HSEs that reside within annotated Helitrons (caenorhabditis.org) (*Supplementary file 6*). All examined *Caenorhabditis* species show significant enrichment of HSEs within Helitrons (p<1.0e-03 in all cases, binomial test), even though the percentages of HSEs located in annotated Helitrons range between 0.2% in *C. monodelphis* to 46.2% in *C. elegans* (*Figure 3C*). We also examined these other *Caenorhabditis* genomes for HSE clustering. Of the 27 species examined, nine species have at least ten 1 kb windows with more than 10 HSEs (*Figure 3C* and *Supplementary file 6*). In addition to *C. elegans*, the genomes of *C. briggsae, C. brenneri,* and *C. virilis* each contain more than 100 such clusters of Helitron-embedded HSEs (*Figure 3C*), and several genomes contain at least one cluster with more than 50 HSEs in a 1 kb window (*Supplementary file 6*). These results demonstrate that the strong association of HSEs with Helitrons, including high-density clustering in some species, is widespread in the *Caenorhabditis* genus.

## Distinct genes with Helitron-provided HSEs are part of the heat shock response in *C. elegans* and *C. briggsae*

Our finding that Hac-HSEs are present in all other available *Caenorhabditis* species suggested that Helitrons may have been co-opted to drive gene expression during HS outside of *C. elegans*. To test this hypothesis, transcriptional profiling was performed in L2 stage *C. briggsae* worms during NHS and HS conditions, and positions of significantly up-regulated genes determined using DESeq2 (*Love et al., 2014*) (fold change >2, adjusted p-value<0.01) were compared to Hac-HSE locations (*Supplementary files 7* and *8*). Helitron positions were obtained from the RepeatMasker track at the UCSC Genome Browser (*Casper et al., 2018*; *Kent et al., 2002*) (genome build cb4), and HSEs were identified using the MEME-derived HSE sequence obtained from *C. elegans* ChIP-seq data shown in *Figure 1B*. For reference, the same analyses were performed using our *C. elegans* expression data from *Figure 2*. Notably, without the additional ChIP-seq data to identify direct HSF-1 binding, we observe many more HS up-regulated genes associated with Hin-HSEs in *C. elegans* (*Figure 2B* versus *Figure 4A*), suggesting this type of analysis may overestimate the true Hin-HSE targets of HSF-1 in the genome. In both *C. elegans* (*Figure 4A*) and *C. briggsae* (*Figure 4B*), we observe significant enrichment of HS up-regulated genes within 2.5 kb of Hac-HSEs compared to background levels observed by chance (*C. elegans*, p<1.0e-15; *C. briggsae*, p=1.8e-05, $\chi^2$ test). Similar results were obtained for Hin-HSE genes in *C. elegans* (*Figure 4A*) and *C. briggsae* (*Figure 4B*). These results suggest that Hac-HSEs contribute to the HSR in *C. briggsae* in addition to *C. elegans*.

*C. elegans* and *C. briggsae* are estimated to have diverged less than 30 million years ago (*Cutter, 2015*). If Helitron elements have propagated since this divergence, distinct genes may have been integrated into the HSR by species-specific Helitron insertions in these two species. To analyze potential differences in HSR genes in *C. elegans* and *C. briggsae*, we first identified orthologs for HSR genes between the two species using ParaSite (*Howe et al., 2017*; *Howe et al., 2016*) (genomes ce11 and cb4) (*Supplementary file 8*). When we compared HSR genes that have upstream Hac-HSEs with their orthologs in the other species, there was only one ortholog pair up-

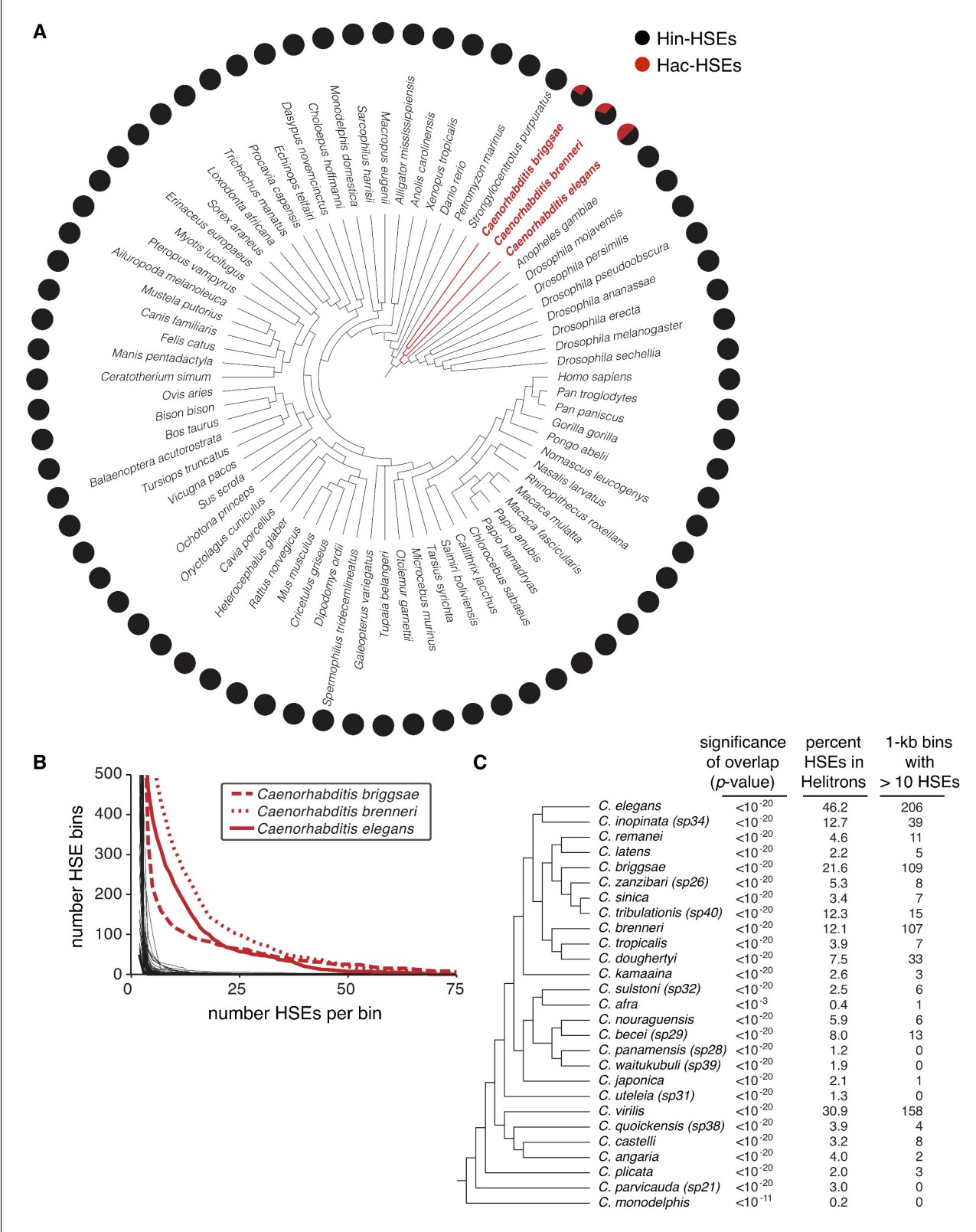

**Figure 3.** Helitrons contribute significant numbers of HSEs to other *Caenorhabditis* genomes. (**A**) Phylogeny of all metazoan genomes hosted at the UCSC genome browser (genome.ucsc.edu) that have RepeatMasker-generated annotations of repetitive elements (see *Supplementary file 5* for genome assembly IDs). Adjacent pie charts display the proportion of total HSEs that reside within (red) or outside of (gray) annotated Helitrons. To prevent possible species-specific biases found in HSEs from influencing our results, we scanned for HSEs using the canonical species-independent

*Figure 3 continued on next page*

*Figure 3 continued*

sequence NGAANNTCCNNGGAN and its reverse complement (*Perisic et al., 1989*). Names of species with significant HSE-Helitron overlap (cumulative binomial distribution with p-value<0.001, see *Supplementary file 5*) are highlighted in red. (B) Cumulative distribution analysis of the density of HSEs for the genomes shown in panel A. The number of HSEs per 1 kb window (bin) in each genome was determined. Curves are plotted as the number of bins with a value that is greater than or equal to the indicated numbers of HSEs per bin. The three *Caenorhabditis* genomes that show numerous 1 kb windows with large numbers of HSEs per bin are highlighted in red. (C) Phylogeny of species hosted at the *Caenorhabditis* Genomes Project (http://www.caenorhabditis.org) that have RepeatMasker-generated annotations of repetitive elements (see *Supplementary file 6* for genome assembly IDs). Next to each species name is the statistical significance of HSE/Helitron overlap (cumulative binomial probability), proportion of total HSEs found to overlap with Helitrons, and number of 1 kb windows with greater than 10 HSEs per window. Complete data are found in *Supplementary file 6*.

regulated in both *C. elegans* and *C. briggsae* that also has upstream Hac-HSEs in both species (designated as 'shared') and this one gene appears to encode for a full-length Helitron transposase (see below). In contrast, the remaining orthologous pairs have Hac-HSEs in one species but not the other (designated as 'unique') (*C. elegans*, n = 38; *C. briggsae*, n = 26) (*Figure 4C*). Thus, many of the genes in the unique categories of Hac-HSE-containing up-regulated orthologs in *C. elegans* and *C. briggsae* seem to be newly responsive to HS via promoter-associated Hac-HSEs. Indeed, of the 38 unique Hac-HSE-containing *C. elegans* orthologs, only three are also up-regulated during heat shock in *C. briggsae*, and one of these has upstream Hin-HSEs. Likewise, of the 26 unique *C. briggsae* orthologs, eight are also up-regulated during heat shock in *C. elegans*, and five of these have Hin-HSEs. This contrasts from up-regulated orthologs with Hin-HSEs, where 27 up-regulated genes shared between species also have Hin-HSEs in both species. In agreement with many of these shared Hin-HSE HSR orthologs being conserved HSF-1 targets, over half (15/27, or 56%) of these genes are bound by HSF-1 in *C. elegans* (*Supplementary file 4*). Thus, differences in Hac-HSEs between *C. elegans* and *C. briggsae* orthologs appear to result in species-specific gene expression upon heat shock, while Hin-HSEs drive a more conserved HSR among orthologs between the two species.

To illustrate how Hac-HSEs are associated with differences in heat shock responsiveness of otherwise orthologous genes in *C. elegans* and *C. briggsae*, we present individual cases in *Figure 4D–I*. As an example of a unique Hac-HSE gene in *C. elegans*, *col-52* has 86 *HelitronY4_CE*-provided HSEs in its upstream promoter region (*Figure 4D*) and is up-regulated during heat shock in *C. elegans* (*Figure 4E*). In contrast, in *C. briggsae* this same gene lacks Hac-HSEs (*Figure 4D*) and is non-responsive to HS (*Figure 4E*). Similarly, the *C. briggsae* gene *CBG00919* has 12 Hac-HSEs provided by *Helitron7_CB* (*Figure 4F*) and is up-regulated during heat shock in *C. briggsae* (*Figure 4G*), while the *C. elegans* ortholog *C01G6.3* lacks Hac-HSEs (*Figure 4F*) and is not up-regulated during heat shock (*Figure 4G*). Interestingly, the single shared up-regulated ortholog with Hac-HSEs in both *C. elegans* and *C. briggsae* seems to encode for a full-length Helitron transposase. In *C. elegans*, the Rep-Helicase-encoding gene *F33H12.6* resides within an annotated full-length copy of *Helitron1_CE*, has 14 Hac-HSEs in its promoter region (*Figure 4H*), and is up-regulated during heat shock (*Figure 4I*). Likewise, the *C. briggsae* Rep-Helicase ortholog *CBG26851* (determined using BLASTP) (*Altschul et al., 1990*) has 85 upstream Hac-HSEs provided by *Helitron7_CB* (*Figure 4H*) and is up-regulated during heat shock (*Figure 4I*). Currently, it is unclear if this one shared Hac-HSE example derived from vertical inheritance of an HSE-containing Helitron or arose from transposition of a Helitron after the divergence of *C. elegans* and *C. briggsae*. Overall, these observations are consistent with Helitrons mobilizing since the divergence of *C. elegans* and *C. briggsae* and independently incorporating genes into the HSR.

If the independent mobilization of Helitrons in *C. elegans* and *C. briggsae* has led to distinct species-specific genes being incorporated into the HSR, then the types of Hac-HSE-containing genes up-regulated during heat shock might be expected to differ between species. To test this, we performed gene ontology (GO) analysis of the 'unique' Hac-HSE up-regulated orthologs described in *Figure 4C*. Consistent with the GO analysis depicted in *Figure 2I*, we found that unique up-regulated Hac-HSE-containing orthologs in *C. elegans* are enriched for GO terms associated with collagens and GPCR signaling (*Figure 4J*). In contrast, unique up-regulated Hac-HSE orthologs in *C. briggsae* are associated with carbohydrate metabolism (*Figure 4J*). Although many HS-induced Hin-HSE orthologs were not shared between *C. elegans* and *C. briggsae*, the GO term associated with phosphorylation is enriched in the unique Hin-HSE genes from both species (*Figure 4J*). As

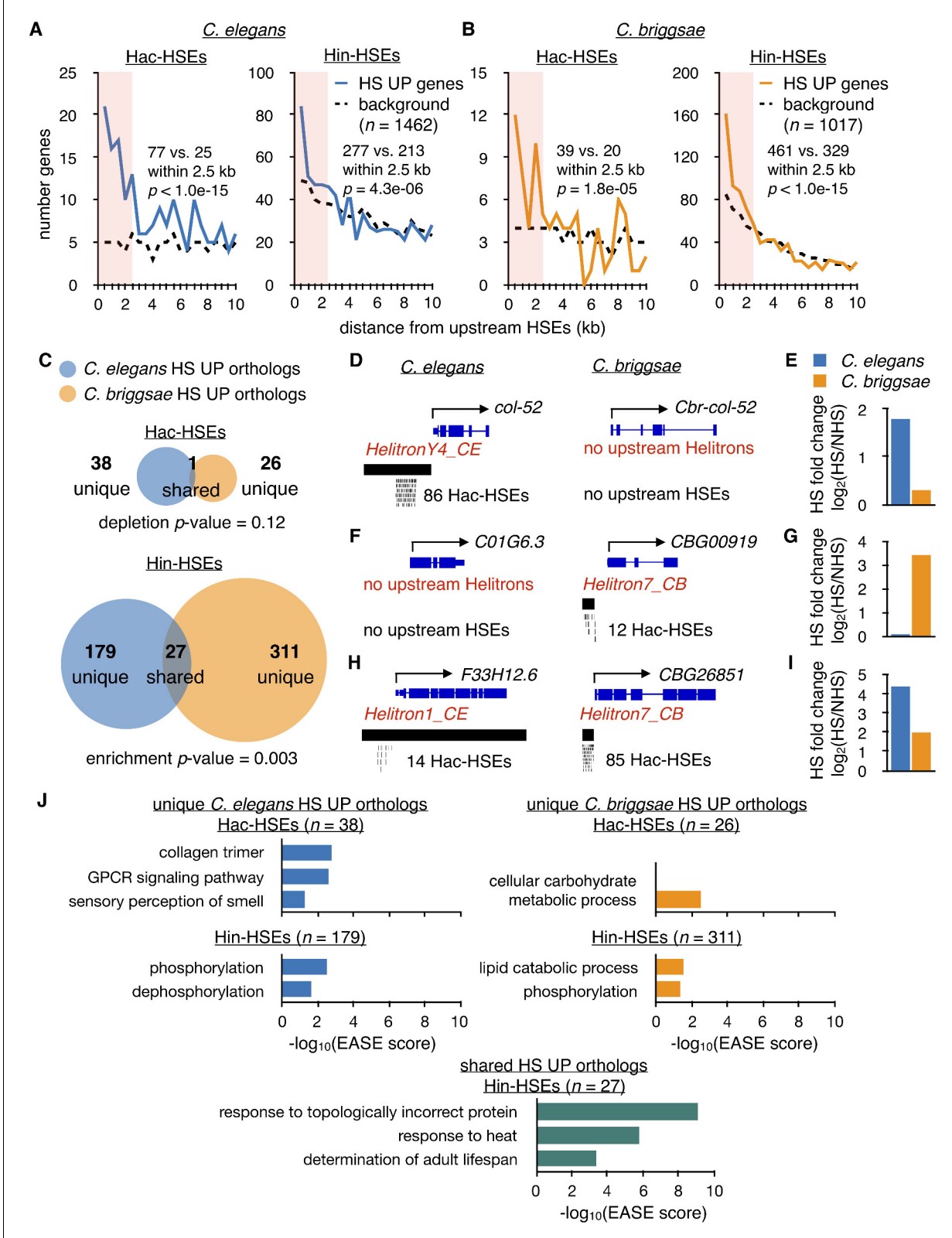

**Figure 4.** Hac-HSEs are associated with distinct gene sets up-regulated by HS in *C. elegans* and *C. briggsae*. (**A**) Number of *C. elegans* HS UP genes (*n* = 1462) within 500 bp bins vs. distance from Hac- and Hin-HSEs. HS UP genes are defined as those with a fold change after HS >2 and an adjusted p-value<0.01 determined using DESeq2 (*Love et al., 2014*). Solid blue lines represent numbers of HS UP genes observed, while black dashed lines represent background numbers resulting from chance. Seventy-seven HS UP vs. 25 expected genes have Hac-HSEs within 2.5 kb upstream, resulting in

*Figure 4 continued*

significant enrichment (p<1.0e-15, $\chi^2$ test). Two hundred and seventy-seven HS UP vs. 213 expected genes have Hin-HSEs within 2.5 kb upstream, resulting in significant enrichment (p=4.3 e-06, $\chi^2$ test) (B) Number of *C. briggsae* HS UP genes (n = 1017) within 500 bp bins vs. distance from Hac- and Hin-HSEs. Solid orange lines represent numbers of HS UP genes observed, while dashed black lines represent background numbers resulting from chance. Thirty-nine HS UP vs. 20 background genes have Hac-HSEs within 2.5 kb upstream, resulting in significant enrichment (p=1.8e-05, $\chi^2$ test). Four hundred and sixty-one HS UP vs. 329 background genes have Hin-HSEs within 2.5 kb upstream, resulting in significant enrichment (p<1.0e-15, $\chi^2$ test). (C) Venn diagrams showing overlap of orthologous HS UP genes in *C. elegans* (blue) and *C. briggsae* (orange) that have Hac- or Hin-HSEs within 2.5 kb upstream in both species. Unique HS UP orthologs have upstream HSEs in either *C. elegans* or *C. briggsae*, while shared HS UP orthologs have upstream HSEs in both species. There is significant enrichment in the overlap of *C. elegans* Hin-HSE genes with *C. briggsae* orthologs up-regulated after HS (p=0.003, hypergeometric test). (D) Genome browser screenshots of *C. elegans col-52* and its *C. briggsae* ortholog *Cbr-col-52*. *C. elegans col-52* has 86 upstream Hac-HSEs supplied by a copy of *HelitronY4_CE* and *Cbr-col-52* lacks upstream Hac-HSEs. (E) HS fold-change values for *C. elegans col-52* (blue) (3.5 fold) and *C. briggsae Cbr-col-52* (orange) (1.2 fold) determined using DESeq2. (F) Genome browser screenshots of *C. elegans C01G6.3* and its *C. briggsae* ortholog *CBG00919*. *C01G6.3* lacks upstream Hac-HSEs and *CBG00919* has 12 Hac-HSEs provided by a copy of *Helitron7_CB*. (G) HS fold-change values for *C. elegans C01G6.3* (blue) (1.1 fold) and *C. briggsae CBG00919* (orange) (11 fold) determined using DESeq2. (H) Genome browser screenshots of *C. elegans F33H12.6* and its *C. briggsae* ortholog *CBG26851*. *F33H12.6* resides within a full-length copy of *Helitron1_CE* and has 14 upstream Hac-HSEs. *CBG26851* has 85 upstream Hac-HSEs supplied by a copy of *Helitron7_CB*. (I) HS fold-change values for *C. elegans F33H12.6* (blue) (21 fold) and *C. briggsae CBG26851* (orange) (4.0 fold) determined using DESeq2. (J) Significantly-enriched gene ontology (GO) terms identified using DAVID (*Huang et al., 2009a*; *Huang et al., 2009b*) for unique Hac- and Hin-HSE up-regulated orthologs in *C. elegans* (blue) and *C. briggsae* (orange), as well as shared Hin-HSE up-regulated orthologs (aquamarine).

expected, shared Hin-HSE orthologs are enriched for terms associated with canonical HSR targets, such as response to topologically incorrect protein, response to heat, and the determination of adult lifespan (*Figure 4J*). Overall, these results provide further evidence that the independent mobilization of Helitrons in *C. elegans* and *C. briggsae* has resulted in the incorporation of unique genes into the HSR.

## HSEs provided by Helitrons diversify the heat shock response in *C. elegans*

We next asked if re-wiring of the HSR by Helitrons might have occurred in *C. elegans* at the population level. Wild isolates of *C. elegans* were screened for the absence of Helitrons relative to the N2 reference strain by using whole-genome sequencing (WGS) data available from CeNDR (*Cook et al., 2017*) in conjunction with the structural variant caller Delly (*Rausch et al., 2012*). From this, HSE-containing Helitrons present in the promoters of three different N2 genes were confirmed to be absent upstream of the same genes in the wild isolates QX1211 and MY16. In the N2 strain, *HelitronY4_CE* sequence provides 96 Hac-HSEs to *nhr-247* (*Figure 5A*), *HelitronY1A_CE* provides 30 Hac-HSEs to *str-96* (*Figure 5B*), and *Helitron2_CE* provides 9 Hac-HSEs to *fbxa-102* (*Figure 5C*). All of these genes associate with HSF-1 and Pol II and increase in expression during HS in N2 (*Figure 5A–C* and *Supplementary files 2* and *3*). The Hac-HSEs are the only HSEs within the putative promoter regions of these N2 HSR genes, arguing that their ability to be up-regulated during heat shock is largely dependent upon the presence of Helitrons. As Helitrons are thought to be replicated in place before transposition (*Kapitonov and Jurka, 2001*), loci lacking Helitrons in wild isolates likely represent the ancestral pre-transposition state. These results suggest that Helitrons have recently mobilized to new regions in *C. elegans* and are actively diversifying the HSR within the species.

We then compared the heat shock response of *nhr-247*, *str-96*, and *fbxa-102* in N2 versus the wild isolates that lack Hac-HSEs associated with these genes. While these three genes were strongly up-regulated during HS versus non-HS conditions in N2 (*nhr-247*, 254 fold, p=0.01; *str-96*, 7.0 fold, p=0.006; *fbxa-102*, 8.3 fold, p=0.03; Welch Two Sample *t*-test), their induction was greatly reduced in QX1211 (*nhr-247*, 3.2 fold, p=0.16; *str-96*, 3.6 fold, p=0.07; Welch Two Sample *t*-test) and MY16 (*fbxa-102*, 0.4 fold, p=0.29; Welch Two Sample *t*-test) (*Figure 5D–E*). The canonical HSR gene *hsp-70* showed similar levels of HS induction in the wild isolates compared to N2, ruling out that a generally attenuated HSR is responsible for the diminished induction of *nhr-247* and *str-96* in QX1211 and *fbxa-102* in MY16 (*Figure 5D–E*). Additionally, the Helitron Rep-Helicase transposase genes found in full-length copies of *Helitron1_CE* also exhibited similar levels of HS up-regulation in the wild isolates compared to N2, showing that the Helitron HS response is intact in these strains (*Figure 5D–E*). Altogether, these results demonstrate that Helitrons can provide functional HSEs to adjacent genes

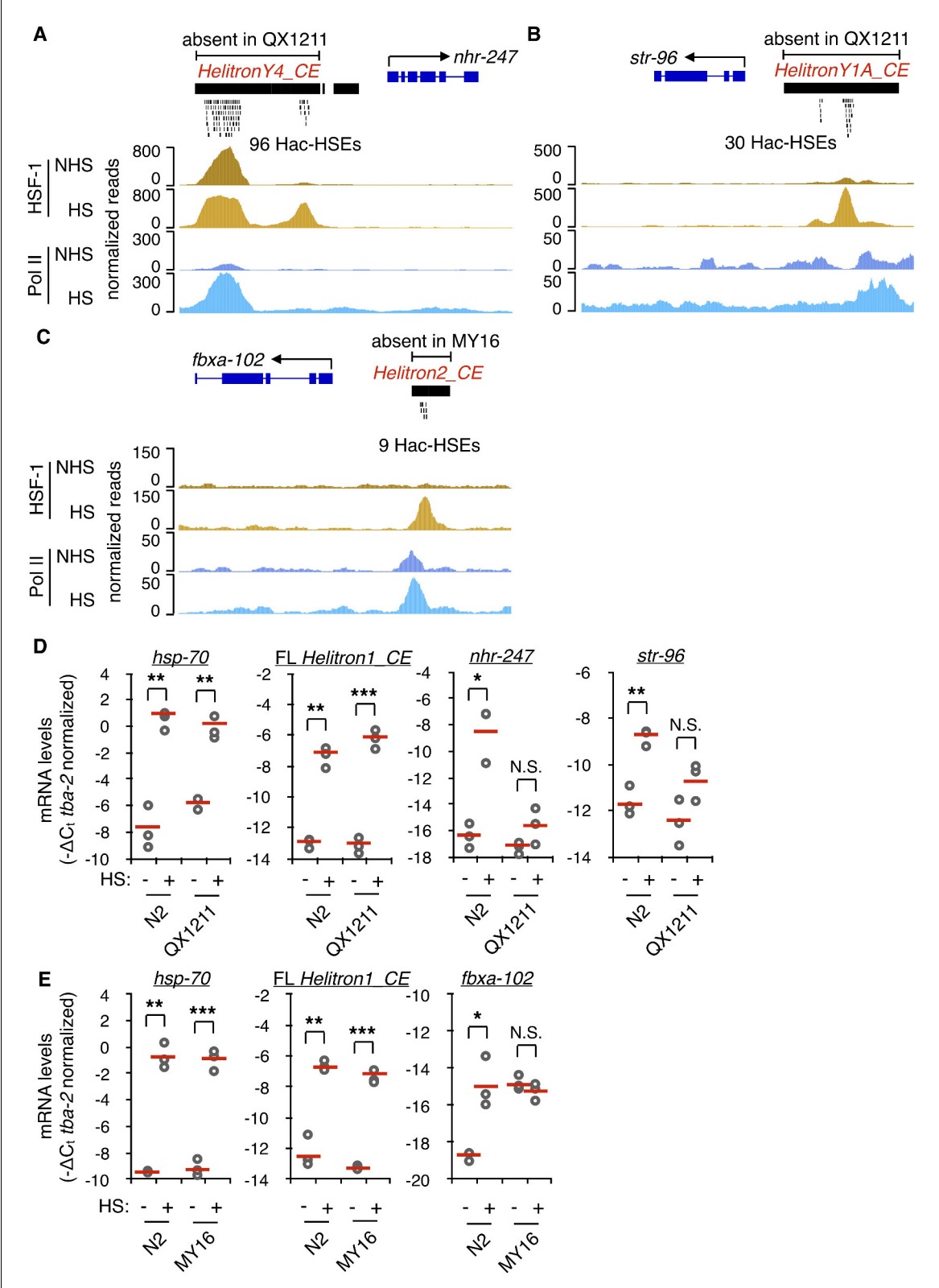

**Figure 5.** Hac-HSEs diversify the HSR in *C. elegans*. (**A**) Genome browser screenshot depicting the region surrounding the Hac-HSE HSR gene *nhr-247* showing normalized ChIP-seq signal for HSF-1 and Pol II under NHS and HS conditions (ChIP-seq data obtained from ***Li et al., 2016***). *HelitronY4_CE* sequence containing 96 Hac-HSEs is located upstream of *nhr-247* in the wild-type reference strain N2, but is absent in the wild isolate QX1211. (**B**) Genome browser screenshot depicting the region surrounding the Hac-HSE HSR gene *str-96* showing normalized ChIP-seq signal for HSF-1 and Pol II
*Figure 5 continued on next page*

*Figure 5 continued*

under NHS and HS conditions. *HelitronY1A_CE* sequence containing 30 Hac-HSEs is located upstream of *str-96* in the wild-type reference strain N2, but is absent in the wild isolate QX1211. (**C**) Genome browser screenshot depicting the region surrounding the Hac-HSE HSR gene *fbxa-102* showing normalized ChIP-seq signal for HSF-1 and Pol II under NHS and HS conditions. *Helitron2_CE* sequence containing nine Hac-HSEs is located upstream of *fbxa-102* in the wild-type reference strain N2, but is absent in the wild isolate MY16. (**D**) Scatterplots showing relative mRNA expression levels (-$\Delta C_t$) determined using RT-qPCR (gray open circles) of *nhr-247* and *str-96* before (NHS) and after heat shock (HS) in N2 worms that have and QX1211 worms that lack upstream Hac-HSEs. Expression levels of the Hin-HSE HSR gene *hsp-70* and *Helitron1_CE* transposase are also shown. RT-qPCR data are normalized to the heat-shock-unchanging housekeeping gene *tba-2*. Data obtained from three independent biological replicates are shown, with their mean values represented using solid red lines. (**E**) Scatterplots showing relative mRNA expression levels (-$\Delta C_t$) determined using RT-qPCR (gray open circles) of *fbxa-102* before (NHS) and after heat shock (HS) in N2 worms that have and MY16 worms that lack upstream Hac-HSEs. Expression levels of the Hin-HSE HSR gene *hsp-70* and *Helitron1_CE* transposase are also shown. RT-qPCR data are normalized to the heat-shock-unchanging housekeeping gene *tba-2*. Data obtained from three independent biological replicates are shown, with their mean values represented using solid red lines. In N2 NHS worms, *fbxa-102* was undetectable in one biological replicate. For panels D-E, a single asterisk represents p<0.05, double asterisks represent p<0.01, and triple asterisks represent p<0.001 (Welch Two Sample *t*-test). For all genes with the exception of *Helitron1_CE* transposase, one primer in each qPCR primer pair spans an exon-exon junction to ensure detection of mature mRNA levels.

and suggest that Helitrons are active agents of evolution that alter the HSR within worm populations.

## Discussion

Here we show that across the genomes of multiple *Caenorhabditis* species, Helitrons provide substantial numbers of binding motifs (HSEs) for HSF-1, a highly conserved transcriptional regulator of the heat shock response (HSR). Remarkably, we find that almost two-thirds of *C. elegans* HSEs reside within Helitrons. Consistent with Helitron-embedded HSEs being functional, many are bound by HSF-1 and display increased levels of Pol II after HS in *C. elegans* and are enriched near HS-responsive genes in both *C. elegans* and *C. briggsae*. Comparison of Helitron positions between *C. elegans* and *C. briggsae,* as well as between wild isolates of *C. elegans,* showed variability in Helitron position, suggesting that Helitrons have mobilized recently and may still be active. Our work demonstrates that Helitrons have significantly shaped the HSR in *C. elegans* by supplying HSF-1 binding sites to dozens of genes and, hence, rewiring their responsiveness to HS.

### Helitron transposons have rewired the *C. elegans* heat shock response

Helitrons comprise ~2% of the *C. elegans* genome (*Kapitonov and Jurka, 2001*). Of these, there are six full-length autonomous (predicted to encode a transposase and likely replication competent) Helitrons and ~5000 non-autonomous elements that lack protein-coding potential; many of these non-autonomous elements are fragments of full-length autonomous versions (*Hubley et al., 2016*; *Kapitonov and Jurka, 2001*). We have found that approximately 19% of the annotated Helitrons contain at least one HSE motif and that almost two-thirds of all *C. elegans* HSEs reside within Helitrons, many of which exist in high-density clusters. These Helitron-acquired HSEs (Hac-HSEs) are capable of recruiting HSF-1 and promoting the expression of proximal genes in response to HS. Thus, while transposase-encoding Helitrons may have initially benefitted from HSEs in their promoter as a way to mobilize upon cellular stress, the widespread Helitron-mediated insertion of HSEs within the *C. elegans* genome now provides heat-shock inducibility to downstream genes. In comparison to Helitron-independent (Hin-HSEs) HSR genes, Hac-HSE associated genes tend to exhibit similar fold changes in HS induction but relatively lower levels in mRNA expression and Pol II occupancy under non-heat-shock conditions. This difference may be attributed to transposon silencing mechanisms, including the formation of repressive chromatin at and around Helitrons (*Slotkin and Martienssen, 2007*). Consistent with chromatin-based silencing of Helitrons affecting the expression of nearby genes, HSR genes proximal to Helitrons display higher levels of repressive histone marks during non-HS conditions compared to those more distant. It is possible that the high number of HSEs provided by Helitrons helps overcome this repressive landscape by presenting multiple opportunities for HSF-1 binding and transcriptional induction of nearby genes. The potential for Helitrons to trigger silencing of adjacent genes but also enable them to respond to HS further supports a role for these TEs in rewiring gene expression patterns in *C. elegans*.

Mobilization of TEs is inherently mutagenic and the consequence can be deleterious, beneficial, or selectively neutral depending on the insertion position and the sequence provided (*Sundaram and Wang, 2018*). While it remains to be determined whether Hac-HSEs provide a fitness advantage to the organism by promoting gene expression during HS, several lines of evidence suggest this is possible. Some Hac-HSE genes are associated with phenotypes indicative of roles in stress, show signatures of selection, and share specific GO terms. Importantly, we observe no overlap in the GO term enrichments for Hac-HSE HSR genes versus all Helitron-associated genes. This argues against sequence-biased mechanisms of Helitron insertion as a predominant explanation for the specific GO categories associated with HS-induced Hac-HSE genes. Specifically, genes encoding collagens and G protein-coupled receptors (GPCRs) are enriched in the Hac-HSE class. Both collagens and GPCRs are among the largest gene families in *C. elegans* (*Johnstone, 2000*; *Thomas and Robertson, 2008*), potentially providing a rich environment for Helitron-mediated functional divergence to occur. Consistent with roles in the HSR, collagen genes have been previously reported to be up-regulated during heat stress in chicken cells (*Sun et al., 2015*) and human cells (*Li et al., 2018*), as well as in *C. elegans* (*Brunquell et al., 2018*; *Schreiner et al., 2019*), suggesting that increasing collagen output is a conserved component of the HSR. Likewise, GPCRs are transmembrane receptors with roles in neuronal signaling (*Thomas and Robertson, 2008*), including the induction of canonical HSF-1 targets in non-neuronal tissues and, hence, thermotolerance in *C. elegans* (*Maman et al., 2013*). An alternative explanation for the enrichment of collagen and GPCR genes associated with Hac-HSEs is that Helitron promoter insertions are better tolerated for these gene families. Further studies will be needed to determine whether recruitment of these specific collagen and GPCR genes into the HSR is neutral, deleterious, or enhances the ability of *C. elegans* to withstand or recover from heat stress.

## Helitrons have shaped the heat shock response in multiple *Caenorhabditis* species

Roughly half of all up-regulated HSF-1 targets observed in this study have Hac-HSEs, demonstrating that Helitrons have had a significant role in shaping the *C. elegans* HSR. Beyond *C. elegans*, we observed a significant association of HSEs with Helitrons in all examined *Caenorhabditis* species, suggesting that this is a long-standing phenomenon. In contrast, over a broad range of bilateria species for which Helitrons have been identified, we detected no increased correlation between HSEs and Helitrons and no high-density clusters of HSEs. The presence of Hac-HSEs across *Caenorhabditids* led us to ask if Helitrons might be associated with differences in heat-shock-responsive genes within and between species. Similar to *C. elegans*, we found a number of Hac-HSEs upstream of HSR genes in *C. briggsae*. However, with the exception of what appears to be an incompletely annotated autonomous Helitron in *C. briggsae*, none of these HSR genes had upstream Hac-HSEs present in both species. Indeed, we find that in most cases, these orthologs are uniquely heat-shock responsive in the species with upstream Hac-HSEs and lack heat-shock responsiveness in the species without upstream Hac-HSEs. This difference in gene sets that respond to HS is consistent with previous studies that concluded that substantial Developmental System Drift contributes to the divergence in genetic networks associated with the phenotypically very similar *C. elegans* and *C. briggsae* (*Verster et al., 2014*). In some cases, the evolution of distinct functions for seemingly orthologous genes may be driven by changes in the regulation of their expression (*Halfon, 2017*). Indeed, it appears that with a single Helitron insertion, new genes have been incorporated into the HSR in a species-specific way. Remarkably, we find this rewiring occurs even between diverse *C. elegans* wild isolates, underscoring the role that TEs may play in providing variation in stress responses within different populations of a species. Intriguingly, it was recently reported that two different isolates of *C. briggsae* (AF16 and HK104) exhibit substantial gene expression differences in response to thermal stress (*Mark et al., 2019*). It will be interesting to examine if any of these differences might be associated with strain-specific Helitron insertions.

## Helitrons expand the landscape of HSF-1 targets

The impact of Helitron-provided HSEs on gene expression may extend well beyond the HSR. HSF-1 has roles in counteracting other types of stress that result in protein unfolding and aggregation, such as exposure to heavy metals or azide (*Labbadia and Morimoto, 2015*). In addition to stress

responses, HSF-1 has been shown to play essential roles in developmental processes in a variety of organisms, including mice, flies and worms (*Christians et al., 2000*; *Jedlicka et al., 1997*; *Li et al., 2016*; *Santos and Saraiva, 2004*; *Takaki et al., 2006*). Furthermore, HSF-1 has been found to support pro-growth transcriptional programs in yeast, worms, and human cancer cells (*Li et al., 2016*; *Mendillo et al., 2012*; *Solís et al., 2016*), suggesting commonalities among non-canonical HSF-1 targets across species (*Li et al., 2017*). In this study, we increase the number of identified HSF-1 HS-targets from 60 to 119 by considering HSF-1 binding that overlaps with HSE-containing repetitive elements that are normally excluded. Given their prevalence in the *C. elegans* genome, Helitron-provided HSEs have the potential to shape gene expression programs beyond the HSR, such as during other environmental challenges as well as development.

Stress responses, by their nature, allow organisms to adapt to new environments. As a result, it may be evolutionarily advantageous to rapidly rewire genes into stress response pathways. Although we speculate that the HSR was initially co-opted as a selfish way to promote mobilization of parasitic Helitrons upon cellular stress, Hac-HSEs may also provide more genetic variation and opportunities for positive and negative selective pressures to alter the host response to heat shock. By leveraging HSEs found in Helitrons that litter their genomes, *Caenorhabditis* species might be able to harness a single TE-mediated evolutionary innovation rather than needing to evolve HSEs by multiple single-nucleotide changes. We predict that this rewiring of the genes controlled by the otherwise highly conserved heat shock response between and among different *Caenorhaditis* species has the potential to impact *Caenorhabditis* fitness and adaptation to novel stress conditions.

# Materials and methods

## Key resources table

| Reagent type (species) or resource | Designation | Source or reference | Identifiers | Additional information |
|---|---|---|---|---|
| Genetic reagent (*C. elegans*) | N2 | | | wild-type strain |
| Genetic reagent (*C. elegans*) | QX1211 | | | Wild isolate |
| Genetic reagent (*C. elegans*) | MY16 | | | Wild isolate |
| Genetic reagent (*C. briggsae*) | AF16 | | | wild-type strain |
| Commercial assay or kit | Illumina TruSeq Stranded Total RNA Library Prep Gold Kit | Illumina | Illumina: 20020598 | |
| Software, algorithm | FigTree v1.4.4 | http://tree.bio.ed.ac.uk/software/figtree/ | | |
| Software, algorithm | Bowtie2 v2.3.3.1 | *Langmead and Salzberg, 2012* | | Sequence alignment |
| Software, algorithm | MEME v5.0.3 | *Bailey and Elkan, 1994* | | Motif finder |
| Software, algorithm | Tomtom v5.0.2 | *Gupta et al., 2007* | | Motif comparison |
| Software, algorithm | Cistrome | *Liu et al., 2011* | | Generation of heat maps |
| Software, algorithm | DESeq2 v1.18.1 | *Love et al., 2014* | | Differential expression analysis of RNA-seq |
| Software, algorithm | Delly v0.8.1 | *Rausch et al., 2012* | | Genome comparisons |

*Continued on next page*

*Continued*

| Reagent type (species) or resource | Designation | Source or reference | Identifiers | Additional information |
|---|---|---|---|---|
| Software, algorithm | bcftools v1.9 | *Narasimhan et al., 2016* | | Genome comparisons |

## *C. elegans* and *C. briggsae* strains and culture

*C. elegans* and *C. briggsae* were cultured using standard conditions at 20℃ (*Wood, 1988*) unless otherwise noted. The Bristol N2 strain was used as the wild type laboratory strain for *C. elegans* and AF16 for *C. briggsae. C. elegans* wild isolates used are as follows:

QX1211, obtained from an urban garden in San Francisco, CA, USA (37.750198 ˚ latitude −122.433098 ˚ longitude) (sampled and isolated by Matthew Rockman)
MY16, obtained from an urban garden in Mecklenbeck, Germany (51.93 ˚ latitude 7.57 ˚ longitude) (sampled and isolated by Hinrich Schulenberg)

## ChIP-seq data mapping, peak calling, and normalization

*C. elegans* HSF-1 and Pol II ChIP-seq data were obtained from *Li et al. (2016)* (GEO Accession Number GSE81523). Histone modification data were obtained from the modENCODE server. Sequencing reads were aligned to a non-repeat-masked version of the *C. elegans* N2 reference genome (ce11) using Bowtie2 v2.3.3.1 (*Langmead and Salzberg, 2012*) with the command `bowtie2 —-no-unal —-very-sensitive`. HSF-1 peaks present at either 20℃ or 34℃ and their summits were called from Bowtie2-aligned reads with the MACS2 v2.1.1 (*Zhang et al., 2008*) command `macs2 -callpeak -g ce —-keep-dup auto —-call-summits -q 1e-6` using combined biological replicates and the single input replicate available. To normalize HSF-1 and Pol II ChIP-seq data for comparative purposes, Bowtie2-mapped reads from combined biological replicates were filtered for duplicates using `macs2 filterdup -g ce —-keep-dup auto`, and the condition with more mapped reads after filtering was randomly sampled down using `macs2 randsample` so the total number of reads considered were identical between conditions. Finally, pileup of filtered and sampled reads was performed using `macs2 pileup` with the `—-extsize` parameter set to the fragment lengths predicted by MACS2 during peak-calling steps.

## Motif identification and scanning

Motifs enriched in 101 bp non-repeat-overlapping HSF-1 summit regions were identified using MEME v5.0.3 (*Bailey and Elkan, 1994*) using the command `meme -mod zoops -dna -revcomp -markov_order 1`. The most significant motif identified in HSF-1 peak summits closely resembles the previously-identified HSE motif using the same dataset (*Li et al., 2016*). To identify motifs present in HSF-1 summit regions overlapping with Helitrons, 101 bp regions centered around HSF-1 peak summits that entirely overlap with annotated Helitrons were also analyzed with MEME. All displayed HSE-like motifs found in Helitron-overlapping HSF-1 peak summits were within the top 10 motifs identified and significantly similar (q < 0.05) to the HSE motif present in non-repeat-overlapping summits determined through the use of Tomtom v5.0.2 (*Gupta et al., 2007*) with the command `tomtom -dist pearson -thresh 0.05 -min-overlap 15`. To scan entire *C. elegans* (ce11) and *C. briggsae* (cb4) genomes for HSE locations, MEME-derived output for HSEs obtained from non-repeat-overlapping *C. elegans* HSF-1 peak summits was used with FIMO v5.0.2 (*Grant et al., 2011*) and the command `fimo —-qv-thresh —-thresh 5e-2`, which reports identified HSEs with q < 0.05. In *C. elegans*, HSF-1-bound HSEs were defined as those that have at least 14 bp overlap with 101 bp HSF-1 summits. To determine whether the clustering of HSEs could arise by chance, a permutation scan of HSEs was performed by randomly selecting the positions of 18,766 (same number as genome HSEs) 15-mers from the genome, and then looking for pile up by counting their numbers in 1 kb bins. While there are 310 bins with greater than 10 HSEs, there are 0 bins with greater than 10 randomly sampled 15-mers. This results in a significant difference between the two (p<1.0e-15, Fisher's exact test). To identify other transcription factor (TF) binding motifs present in Helitrons, experimentally determined motifs for 129 *C. elegans* TFs (*Narasimhan et al., 2015*) as well as DAF-

16 (*Matys et al., 2006*; *Weirauch et al., 2014*) were used in conjunction with FIMO to identify all genomic motifs with p<1.0e-04.

## Calculation of Tajima's D scores in *C. elegans* and *C. briggsae*

Genome-wide Tajima's D scores were calculated over 10 kb sliding windows with a 100 bp step size using VCF-kit (*Cook and Andersen, 2017*). Variation data for 330 *C. elegans* wild isolates were obtained from CeNDR (*Cook et al., 2017*), and variation data for 37 *C. briggsae* wild isolates were previously published (*Thomas et al., 2015*).

## Heatmaps, gene plots, and boxplots

Heatmaps were generated using the heatmap tool found at the Cistrome web server (*Liu et al., 2011*). Boxplots were generated using the `boxplot` function in R v3.5.1 with outliers removed, and notches represent 95% confidence intervals of the median (*R Development Core Team, 2018*). Gene plots were generated by determining the number of HSR genes present within 500 bp bins relative to annotated transcriptional start sites and Hac-HSEs. Background levels resulting from chance were determined using R to sample identical numbers of genes and determining their proximity to Hac-HSEs; this process was repeated 100 times, and average numbers were plotted.

## Heat-shock assays

Heat shock of staged *C. elegans* L2s was performed as described in *Li et al. (2016)* so that our HS RNA-seq experiments could be directly compared to their HSF-1 and Pol II ChIP-seq data. Briefly, starved L1s obtained from bleached embryos were placed on NGM plates and grown at 20°C for 20 hr until mid-L2 stage. Heat-shocked worms were incubated in a 34 ± 0.5°C water bath while non-heat-shocked control worms were kept at 20°C for 30 min before being washed twice in M9 buffer and snap freezing in a dry ice/ethanol bath. Frozen L2 worms were stored at −80°C until total RNA extraction. *C. briggsae* worms were treated in an identical manner for all steps.

## RNA-seq library preparation, sequencing, mapping, and analysis

Total RNA was extracted from frozen L2s using TRIzol reagent and standard protocols that including DNase treatment. 1 µg of extracted total RNA for heat-stressed and control worms (*C. elegans* and *C. briggsae*) was processed using the Illumina TruSeq Stranded Total RNA Library Prep Gold Kit for each sample, with three biological replicates for NHS and HS conditions. Libraries were sequenced using an Illumina HiSeq 4000 sequencer with a run type of SR75. Sequenced reads were mapped to *C. elegans* and *C. briggsae* mRNA transcriptomes using Salmon v0.13.1 (*Patro et al., 2017*) with default parameters and automatic library type detection enabled (`−libType A`), which correctly determined a library type of `SR`. Up-regulated genes in HS-treated versus untreated samples were determined using DESeq2 v1.18.1 (*Love et al., 2014*) with default parameters and a parametric fit type. Significantly up-regulated genes during heat shock were defined as those with $\log_2$(fold change)>1 (fold change >2) and an adjusted p-value<0.01.

## RT-qPCR assays

Total RNA was extracted from frozen L2s using TRIzol reagent and standard protocols that included DNase treatment. 1–2 µg of total RNA was used to prepare cDNA using the Invitrogen cDNA prep kit. 16–40 ng of cDNA was used for each qPCR assay, using PowerUp SYBR Green Master Mix and a QuantStudio machine (ABI Biosystems). All qPCR experiments consisted of three technical replicates for each of three biological replicates. Experimental $C_t$ values were normalized to those of *tba-2*. Sequences for qPCR primers used are as follows:

*tba-2* F: 5'-GCGCCTTCATGGTCGATAA-3'
*tba-2* R: 5'-GAGACAACCTGGGAGATGATTC-3'
*hsp-70* F: 5'-TTCAATGGGAAGGACCTCAAC-3'
*hsp-70* R: 5'-TGGGACAACATCAACGAGTAAA-3'
*Helitron1_CE (Y16E11A.2)* F: 5'-AATCGTCGTGCCAATACCTC-3'
*Helitron1_CE (Y16E11A.2)* R: 5'-GTGCTCACCGAGATGTCTGA-3'
*str-96* F: 5'-GTAAACCGAATGCACATGTAACC-3'
*str-96* R: 5'-TGACTGCGATAACTCCAATATCC-3'

*nhr-247* F: 5'-GTCGGAGCCAATTAGGAGTTT-3'
*nhr-247* R: 5'-TGACTGGGAAATGACTGTGATAA-3'
*fbxa-102* F: 5'-CTCGAGACTAAAGACCGAATGAA-3'
*fbxa-102* R: 5'-GTCCGTAGCAGATTTCAAGACTA-3'

## Identification and verification of Helitron position variability in *C. elegans*

Aligned whole-genome sequencing (WGS) data for N2 and wild isolates were obtained from CeNDR (release 20180527) (*Cook et al., 2017*), and putative deletions in each strain relative to the N2 reference genome were identified using Delly v0.8.1 (*Rausch et al., 2012*) in conjunction with bcftools v1.9 (*Narasimhan et al., 2016*). The Delly `call` command was used to call deletions in each strain individually, and these were merged into a unified site list using the `merge` command. The Delly `call` command was then again used to genotype the merged site list across all strains, and all genotyped strains were merged into a single BCF file using the bcftools `merge` function. The Delly germline filter was then applied to merged genotyped samples using the `filter -f germline` command. The resulting putative germline deletions were intersected with Helitron positions found in the RepeatMasker track available from the UCSC Genome Browser (genome.ucsc.edu). To verify the presence of Helitrons at specific loci in N2 worms or absence in wild isolates, PCR was performed using primers designed to flank regions of interest. Expected amplicon sizes were verified using agarose gel electrophoresis, and sequences were verified through Sanger sequencing. Sequences for genotyping primers are as follows:

DEL.*HelitronY1A.str-96* F: 5'-AGGCGATCTGGGAAATGAG-3'
DEL.*HelitronY1A.str-96* R: 5'-GCATCAGCAAACAGAGGAAG-3'
DEL.*HelitronY4.nhr-247* F: 5'-TCCCAAACAACGCGAAAC-3'
DEL.*HelitronY4.nhr-247* R: 5'-CAGCGAGTGCAACCAATT-3'
DEL.*Helitron2.fbxa-102* F: 5'-GTTCTTCGTTGGGTCATTCG-3'
DEL.*Helitron2.fbxa-102* R: 5'-CTAGCGTCATGGAAAGGAGT-3'

## Identification of Helitron and HSE loci within genomic datasets

Fasta-formatted genome assemblies with reported RepeatMasker files were downloaded from the UC Santa Cruz (UCSC) genome browser (genome.ucsc.edu) and the *Caenorhabditis* Genome Project (CGP) (v1) (http://www.caenorhabditis.org) (*Fierst et al., 2015*; *Kanzaki et al., 2018*; *Mortazavi et al., 2010*; *Slos et al., 2017*; *Stevens et al., 2019*). See *Supplementary files 5* and *6* for genome assembly IDs. To identify Helitron genome coordinates, UCSC- and CGP-provided RepeatMasker files were downloaded and filtered for lines containing the keyword 'helitron'. To identify HSE genome coordinates, a regular expression search was conducted using the degenerate HSE motif, NGAANNTTCNNGAAN, and its reverse complement (*Perisic et al., 1989*).

## Significance of overlap between HSEs and Helitrons

Coordinates of HSEs and Helitrons were used to identify HSEs that entirely fall within a Helitron element. A binomial probability was calculated that estimates that probability of finding a given number of HSEs within the percentage of the genome that is covered by Helitrons. Significance of deviation from that expected distribution, in which the observed number of HSEs within Helitrons exceeds the expected HSE/Helitron overlap, is represented as a *p*-value (p-value=1 cumulative binomial probability).

## HSE pileups across each genome

For all UCSC and CGP genomes, the number of HSEs in a given region (bin) of the genome was determined using a sliding window approach (window size = 1 kb, step size = 1 kb) for all genome contigs greater than 1 kb in size. For each genome, the number of bins with an HSE count greater than or equal to integers from 1 to 100 were tallied. Data are represented as a cumulative distribution curve (*Figure 3B*) or in tabular form (*Figure 3C*, *Supplementary files 5* and *6*).

## Phylogenetic analyses

Newick-formatted phylogenetic trees were generated for UCSC and CGP genomes (see *Supplementary files 5* and *6*) and displayed using FigTree v1.4.4 (http://tree.bio.ed.ac.uk/software/figtree/).

## Acknowledgements

We thank Scott Rifkin for helpful advice, discussions and critical reading of the manuscript. We also thank members of the Pasquinelli laboratory for critical reading of the manuscript and members of the *Caenorhabditis* Genomes Project for pre-publication access to genome data. This work was supported by the National Institutes of Health [R35 GM127012 to AEP, R35 GM133633 to MDD, and T32 GM007240 to BVT], the National Cancer Institute [T32 CA009523 to JMG] and the Pew Biomedical Scholars Program [MDD]. Sequencing of libraries used for RNA-seq analyses was conducted at the IGM Genomics Center, University of California, San Diego, La Jolla, CA. Some strains were provided by the CGC, which is funded by NIH Office of Research Infrastructure Programs (P40 OD010440).

## Additional information

### Funding

| Funder | Grant reference number | Author |
|---|---|---|
| National Institute of General Medical Sciences | R35 GM127012 | Amy E Pasquinelli |
| National Institute of General Medical Sciences | GM133633 | Matthew D Daugherty |
| National Cancer Institute | T32 CA009523 | Jacob M Garrigues |
| National Institute of General Medical Sciences | T32 GM007240 | Brian V Tsu |
| Pew Charitable Trusts | | Matthew D Daugherty |

The funders had no role in study design, data collection and interpretation, or the decision to submit the work for publication.

### Author contributions

Jacob M Garrigues, Conceptualization, Data curation, Software, Formal analysis, Validation, Investigation, Visualization, Methodology; Brian V Tsu, Data curation, Software, Formal analysis, Validation, Investigation, Visualization, Methodology; Matthew D Daugherty, Conceptualization, Formal analysis, Supervision, Funding acquisition, Investigation, Visualization, Methodology, Project administration; Amy E Pasquinelli, Conceptualization, Supervision, Funding acquisition, Project administration

### Author ORCIDs

Matthew D Daugherty (iD) http://orcid.org/0000-0002-4879-9603
Amy E Pasquinelli (iD) https://orcid.org/0000-0002-9511-0039

### Decision letter and Author response

Decision letter https://doi.org/10.7554/eLife.51139.sa1
Author response https://doi.org/10.7554/eLife.51139.sa2

## Additional files

### Supplementary files

• Supplementary file 1. Transcription Factor motifs in Helitrons. Enrichment of Transcription Factor motifs found within *C. elegans* Helitrons compared to genome-wide occurences.

- Supplementary file 2. Heat shock-induced gene expression of L2 stage N2 *C. elegans*. RNA-seq results showing fold change in gene expression of *C. elegans* subjected to heat shock at 34°C versus control animals at 20°C for 30 min.
- Supplementary file 3. *C. elegans* Hin-HSE genes up-regulated by heat shock. List of HSR genes containing Helitron independent HSEs (Hin-HSEs) bound by HSF-1.
- Supplementary file 4. *C. elegans* Hac-HSE genes up-regulated by heat shock. List of HSR genes containing Helitron acquired HSEs (Hac-HSEs) bound by HSF-1.
- Supplementary file 5. Presence of HSEs in Helitrons across species. Analysis of HSEs located within annotated Helitrons for species available at the UCSC Genome Browser.
- Supplementary file 6. Presence of HSEs in Helitrons in *Caenorhabditis* genomes. Analysis of HSEs located within annotated Helitrons for species available through the *Caenorhabditis* Genome Project.
- Supplementary file 7. Heat shock-induced gene expression of L2 stage AF16 *C. briggsae*. RNA-seq results showing fold change in gene expression of *C. briggsae* subjected to heat shock at 34°C versus control animals at 20°C for 30 min.
- Supplementary file 8. Comparison of Hin-HSE and Hac-HSE genes up-regulated by heat shock in *C. elegans* and *C. briggsae*. List of HSR genes containing Helitron independent HSEs (Hin-HSEs) and Helitron acquired (Hac-HSEs) in *C. elegans* and *C. briggsae*.
- Transparent reporting form

## Data availability

The RNA-seq datasets generated in this study are available at the Gene Expression Omnibus (GEO) under accession number GSE135987.

The following dataset was generated:

| Author(s) | Year | Dataset title | Dataset URL | Database and Identifier |
|---|---|---|---|---|
| Garrigues JM, Tsu BV, Daugherty MD, Pasquinelli AE | 2019 | C. elegans and C. briggsae HS RNAseq | https://www.ncbi.nlm.nih.gov/geo/query/acc.cgi?acc=GSE135987 | NCBI Gene Expression Omnibus, GSE135987 |

The following previously published datasets were used:

| Author(s) | Year | Dataset title | Dataset URL | Database and Identifier |
|---|---|---|---|---|
| Contrino S, Smith RN, Butano D, Carr A, Hu F, Lyne R, Rutherford K, Kalderimis A, Sullivan J, Carbon S, Kephart ET, Lloyd P, Stinson EO, Washington NL, Perry MD, Ru-zanov P, Zha Z, Lewis SE, Stein LD, Micklem G | 2012 | H3K27me3 ChIP from L3 stage | http://data.modencode.org/cgi-bin/findFiles.cgi?download=5051 | modMine, modEncode_5051 |
| Cook DE, Zdraljevic S, Roberts JP, Andersen EC | 2017 | All available isotypes | https://elegansvariation.org/data/release/20180527 | Caenorhabditis elegans Natural Diversity Resource, 20180527 |
| Li J, Chauve L, Phelps G, Brielmann RM, Morimoto RI | 2016 | HSF-1 ChIP | https://www.ncbi.nlm.nih.gov/geo/query/acc.cgi?acc=GSE81523 | NCBI Gene Expression Omnibus, GSE81523 |

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
