## [Decision Letter]

**Acceptance summary:**

How transcriptional programs such as those involved in stress response evolve is a major question in biology. Most specifically, understanding how several genes can be recruited to be under the control of a transcription factor so they become co-regulated in response to a particular stimulus remains challenging. Here, the authors show that some transposable elements may carry specific transcription factor binding sites with them while they spread across the genome, allowing at the same time for the recruitment of new genes in heat shock response in nematodes. This is a very elegant study with profound insight on the potential adaptive roles of transposable elements and on major evolutionary changes that can occur quickly in transcriptional networks.

**Decision letter after peer review:**

Thank you for submitting your article "Rewiring of the *Caenorhabditis* heat shock response by helitron transposable elements" for consideration by *eLife*. Your article has been reviewed by three peer reviewers, and the evaluation has been overseen by a Reviewing Editor and Patricia Wittkopp as the Senior Editor. The reviewers have opted to remain anonymous.

The reviewers have discussed the reviews with one another and the Reviewing Editor has drafted this decision to help you prepare a revised submission.

Garrigues and colleagues examine the relationship between the insertion of a transposable element and the heat shock transcriptional network in nematodes. They find that Helitron elements contain Heat Shock Elements that are bound by Heat Shock Factor-1. The authors argue convincingly that genes could be recruited to the heat shock response by insertions of the TE in their vicinity. The reviewers agree that the work is of considerable interest and that it can potentially make an important contribution to the field of genome biology.

Please see the consolidated list of major concerns below. One important point is whether these insertions and the recruitment of new genes under the regulation of HSF-1 could be adaptive or consistent with neutral evolution and potential insertion site biases. Trying to elucidate this aspect would significantly strengthen the manuscript. In addition to the points below, the Introduction appears to be rather short and could be extended in order to put the work in a wider and more general context, for instance the evolution of transcriptional responses, how new genes are recruited to gene expression modules and how TEs could play a role in accelerating the evolution of these networks. This would put the findings in a more generally accessible context and thus pave the way to further work addressing these issues, outside of the specific model studied here.

Essential revisions:

1) Additional experiments are not mandatory but the authors may have access to existing data that may answer this query. Although HSF-1 regulates several key biological functions including, lifespan (Hsu et al., 2003), proteotoxicity (Cohen et al., 2006), development (Barna et al., BMC Dev Biol 2012), and other stresses, the authors have exclusively tested responses to elevated temperatures. Do helitrons also involved in other HSF-1 functions? How Hin-HSEs and Hac-HSEs linked with these other functions? This can be easily tested by exposing N2 and QX1211 worms to the relevant experimental conditions and compare the stress resistance and lifespans of animals of the two strains.

2) A more comprehensive analysis of potential sites within the TEs could be helpful. Other transcription factors such as DAF-16/FOXO are also needed for stress resistance and longevity. Do helitrons also carry DFA-16 recognition sites? This issue is particularly important as the two transcription factors show overlapping functions and both are regulated by the insulin/IGF signaling cascade.

3) Surprisingly, the authors found that the Hac-HSEs regulate very different gene sets in *C. elegans* and in *C. Briggsae* (Figure 4C). This raises the question of what is the significance of the roles played by helitrons in regulating the HSR? Can the authors please comment on that point? Also, it would be good to have some statistical test for whether the overlaps are different between the two species. I understand the argument that Hin-HSE have more conserved functions but it does not seem highly enriched in these data. Likewise, what is the GO term enrichment (Figure 4J) for all Hac or Hin-HSEs vs. the unique ones.

4) This is also another suggestion of experiments which are not mandatory. However, as mentioned above, such data may readily be available.

Are genes that are regulated by Hac-HSE involved in stress resistance and lifespan? This can be addressed by a directed RNAi screen and functional assays. I would suggest to test at least 15 genes to get a sufficient insight.

This should be also tested in other *Caenorhabditis* species.

5) Figure 2H provides a possible explanation to the observation that there is a lower presence of RNA Polymerase II in Hac-HSEs regulated genes (Figure 2G). Performing similar analyses using additional repression and activation epigenetic modifications would further establish this possibility.

6) The authors should not use the word "domestication" to describe how Helitrons could be co-opted or acquired for other functions in the *C. elegans* genome. This word has specific meanings in the evolutionary genetics literature (e.g. a series of robust, controlled, evolutionary changes towards an often human-associated association) that are not supported here. The authors have beautiful data that do not need to use this loaded term.

7) I think that the Discussion is a bit long and parts of it are redundant with or can be included in the Introduction (if necessary). Please tighten both a bit.

8) The authors do a good job of explaining and defending their results in the Discussion and do not good too far in interpretations. One claim that could be softened (slightly) is that selection could be acting on the Hac-HSEs within the *Caenorhabditis* lineage. I agree that it could be, but it could also be drift. The differences in abundance across and within *Caenorhabditis* species suggests that it could be selected because different phylogenetic lineages have similar abundances (with unique complements of genes, probably) but it does not have to be selected in all cases. In *C. elegans*, the authors can look for signatures of selection in and around Hac-HSEs vs. Hin-HSEs (Tajima's D or the like) to look for support. Similar analyses can be done with C. briggsae using the ~40 wild isolates with sequence available. These data will support their claims of selection over drift.

9) While I concur that this conclusion is true at face value, in that helitron insertions modify gene expression patterns when the animals are exposed to heat shock, it is less convincing to me that this has actually led biologically important changes in heat shock response in an adaptive way to benefit the fitness of the organisms. I certainly agree with the idea that helitron insertions (aka mutations) are likely to confer effects that manifest under heat shock; given what is known about the distribution of mutational effects in most organisms, nearly all such mutations will be deleterious or selectively neutral. The polymorphism observed among wild isolates could reflect helitron insertions that ended up being non-deleterious, or weakly deleterious but only under rare conditions of heat shock, thus allowing them to persist and not yet have been weeded out of the population by purifying selection. The fact that comparisons across species show unique effects reinforces the idea that lineage-specific rounds of helitron activity provides a mutational pressure that creates such novel elements with corresponding influence on neighboring genes, but that they usually are evolutionarily short-lived and may not confer any fitness benefit. The above "criticism" aside, the authors are generally quite cautious in not ascribing explicitly a fitness benefit to the helitron-mediated effects on flanking gene expression under heat shock, which I appreciate. Nonetheless, some of the language that the heat shock response is "rewired" leaves implicit the suggestion that this is an adaptive rewiring, which need not be the case. I think it would be valuable to make this point more explicit and more clearly articulate the mutational pressure of helitrons with respect to the potential distribution of fitness effects, with those persisting cases likely being neutral, weakly or conditionally-deleterious, or possibly beneficial; distinguishing among these possibilities would require additional study.

10) It could also be interesting to relate (briefly) the polymorphism among wild isolates to the potential genetic basis to GxE interactions in phenotypes and transcriptome patterns. The enrichment of helitron insertions with specific GO categories is interesting, but could simply reflect sequence-biased mechanisms of insertion. For example, if GPCRs are enriched in certain sequence motifs that are conducive to helitron insertion, then the association of helitron with GPCR could simply be a by-product of the insertional bias. Alternatively, it could be that selection against helitron insertion is weaker for the large multigene family of GPCRs, and so it takes longer for them to be weeded out of the population to allow the helitron-GPCR association to reflect such less negative effects. These kinds of explanations for the GO enrichment analysis would be helpful so as not to mislead readers into thinking that the enrichment must reflect some adaptive story of selective retention in e.g. GPCRS.

---

## [Author Response]

[…] Please see the consolidated list of major concerns below. One important point is whether these insertions and the recruitment of new genes under the regulation of HSF-1 could be adaptive or consistent with neutral evolution and potential insertion site biases. Trying to elucidate this aspect would significantly strengthen the manuscript. In addition to the points below, the Introduction appears to be rather short and could be extended in order to put the work in a wider and more general context, for instance the evolution of transcriptional responses, how new genes are recruited to gene expression modules and how TEs could play a role in accelerating the evolution of these networks. This would put the findings in a more generally accessible context and thus pave the way to further work addressing these issues, outside of the specific model studied here.

As detailed below, we performed new analyses to investigate whether Helitron insertions and the recruitment of new genes under the regulation of HSF-1 could be adaptive or consistent with neutral evolution and potential insertion site biases. While we prefer to remain conservative in our conclusions, we do observe signatures of selection for some Hac-HSE genes, based on Tajima’s D analyses, and the GO term enrichment for Hac-HSE genes is different from that of all Helitron-associated genes, suggesting that insertion bias cannot fully explain the types of genes under the control of Hac-HSEs. Nonetheless we have revised the Discussion to consider alternative possibilities to selection. As suggested, we have also extended the Introduction to include a more thorough background and context for the study.

Essential revisions:1) Additional experiments are not mandatory but the authors may have access to existing data that may answer this query.Although HSF-1 regulates several key biological functions including, lifespan (Hsu et al., 2003), proteotoxicity (Cohen et al., 2006), development (Barna et al., BMC Dev Biol 2012), and other stresses, the authors have exclusively tested responses to elevated temperatures. Do helitrons also involved in other HSF-1 functions? How Hin-HSEs and Hac-HSEs linked with these other functions? This can be easily tested by exposing N2 and QX1211 worms to the relevant experimental conditions and compare the stress resistance and lifespans of animals of the two strains.

As suggested, we analyzed available datasets for phenotypes associated with knockdown or mutation of Hac-HSE genes. That analysis, which is further described in our response to point 4 below, reveals that some Hac-HSE genes have roles in several stress resistance pathways and lifespan, suggesting that they may be important HSF-1 targets. This phenotype information has been added a new column for the Hac-HSE genes in Supplementary file 3.

We restricted our studies to the response of Hac- and Hin-HSE genes to elevated temperature because this is the only condition with available genome-wide HSF-1 binding data (Li et al., 2016). While there is good phenotypic evidence that HSF-1 regulates multiple pathways, the direct transcriptional targets of HSF-1 in these various pathways are yet to be defined at the genome scale through ChIP-seq assays. Additionally, the studies cited above do not include RNA-seq (or microarray) data that could point to specific genes being regulated by the various conditions in an *hsf-1* dependent manner. Hsu et al., 2003, and Cohen et al., 2006, conclude that DAF-16 and HSF-1 are both required for the regulation of aging, including aging-related proteotoxic phenotypes, but only 4 transcriptional targets are identified in Hsu and none in Cohen. See point 2 below for our analysis of possible co-regulation of Hac- and Hin-HSE genes by HSF-1 and DAF-16 in heat shock. The Barna et al., 2012, paper reports an unusual role for HSF-1 in repressing *daf-7* expression to promote dauer formation but does not provide direct evidence that HSF-1 binds to a predicted HSE in this gene. The expression of *daf-7* does not significantly change in our heat shock conditions (see Supplementary file 2).

The question of whether Helitron-provided HSEs contribute to other pathways regulated by HSF-1 is of great interest to us but will require the generation of new HSF-1 ChIP-seq and gene expression datasets under different conditions, which is beyond the scope of this study. The suggestion to expose N2 and QX1211 worms to other experimental conditions and compare lifespans and stress resistance would not allow us to make conclusions about the role of the *nhr-247* and *str-96* genes that have Hac-HSEs in N2 but not QX1211 because these strains are highly diverged from one another, and there are many other polymorphisms between these worms that could contribute to phenotypic differences (Andersen et al., 2012).

2) A more comprehensive analysis of potential sites within the TEs could be helpful. Other transcription factors such as DAF-16/FOXO are also needed for stress resistance and longevity. Do helitrons also carry DFA-16 recognition sites? This issue is particularly important as the two transcription factors show overlapping functions and both are regulated by the insulin/IGF signaling cascade.

To perform a more comprehensive analysis of potential Transcription Factor (TF) sites within Helitrons, we used 129 published *C. elegans* TF binding motifs (Narasimhan et al., 2015) as well as 2 DAF-16 Binding Elements (DBEs) from other sources (Weirauch et al., 2014; Matys et al., 2006) to scan the entire genome for their presence, and then asked whether any had enrichment in Helitrons. While a few other TF motifs were enriched in Helitrons, the most enriched motif in this analysis was for HSF-1 (HSEs), further justifying our focus on HSEs in this study. This analysis also determined that DAF-16 DBEs were significantly depleted from Helitrons. These results have been added as a new supplementary table (Supplementary file 1), and our original supplementary tables have been re-numbered accordingly.

3) Surprisingly, the authors found that the Hac-HSEs regulate very different gene sets in *C. elegans* and in *C. Briggsae* (Figure 4C). This raises the question of what is the significance of the roles played by helitrons in regulating the HSR? Can the authors please comment on that point? Also, it would be good to have some statistical test for whether the overlaps are different between the two species. I understand the argument that Hin-HSE have more conserved functions but it does not seem highly enriched in these data. Likewise, what is the GO term enrichment (Figure 4J) for all Hac or Hin-HSEs vs. the unique ones.

Initially, we were also surprised by this result but think it is consistent with previous studies that concluded that substantial Developmental System Drift (DSD) explains the divergence in genetic networks associated with the phenotypically very similar *C. elegans* and *C. briggsae* (Verster et al., 2014). Since the evolution of distinct functions for seemingly orthologous genes may be driven by changes in the regulation of their expression (Halfon, 2017), we are intrigued by the possibility that Helitrons might contribute to some differences in the genetic networks triggered by HS in *C. elegans* and *C. briggsae.* These points and references are now included in the Discussion.

As requested, we have added the results of hypergeometric tests to determine the significance of overlap to Figure 4. There is significant enrichment of *C. elegans* Hin-HSE genes with *C. briggsae* orthologs up-regulated after HS (*p* = 0.003) but not for Hac-HSE genes (*p* = 0.12). Furthermore, if we consider just high confidence Hin- and Hac-HSE genes bound by HSF-1 based on ChIP data for *C. elegans*, 15/45 Hin-HSE genes are also up-regulated by HS in *C. briggsae* whereas only 1/27 Hac-HSE is also upregulated. Using Fisher’s exact test, these numbers are significantly different (*p* = 0.003) with *C. elegans* Hin-HSE genes enriched for overlap with *C. briggsae* genes, and Hac-HSE genes depleted for overlap in each species.

The GO term enrichments for all Hac- or Hin-HSE genes, shared and unique, are shown in Figure 4J. The exception is the single shared Hac-HSE gene, which seems to be a full-length autonomous Helitron.

4) This is also another suggestion of experiments which are not mandatory. However, as mentioned above, such data may readily be available.Are genes that are regulated by Hac-HSE involved in stress resistance and lifespan? This can be addressed by a directed RNAi screen and functional assays. I would suggest to test at least 15 genes to get a sufficient insight.This should be also tested in other Caenorhabditis species.

Yes, phenotypic data is available for many of the Hac- and Hin-HSE genes. Aggregate data from WormBase (Version WS273) assigns 19/60 Hac-HSE genes with RNAi or genetic mutation phenotypes, some of which are consistent with roles in stress resistance and lifespan (for example, dauer variant, protein aggregation variant, toxin hypersensitive, osmotic stress response variant). For comparison, 31/60 Hin-HSE genes have been associated with RNAi or genetic mutation phenotypes, some of which are consistent with roles in stress resistance and lifespan (for example, dauer variant, protein aggregation variant, toxic chemical response variant, lifespan variant). This phenotype information has been added as new columns for the Hac- and Hin-HSE genes in Supplementary files 3 and 4.

While we would like to perform the suggested RNAi screen to directly test for roles of Hac-HSE genes in heat shock (and possibly in other stress conditions, as well as lifespan control), there are several technical issues that limit the usefulness of RNAi in such a screen. First, many of these genes are neuronal and this tissue is notoriously resistant to RNAi. Second, many of these genes (as well as some of the Hin-HSE genes) are nearly undetectable under NHS conditions, meaning there is little to no template for RNA Dependent Polymerases to amplify the primary RNAi response into the secondary response until HS occurs and the genes are induced; this amplification is important for robust RNAi, and it is unknown if this step functions during HS. Third, several other nematode species do not have the mechanisms found in *C. elegans* that allow for dsRNA uptake by feeding RNAi and spreading throughout the organism, which are important for robust RNAi responses (Nuez and Felix, 2012). While some of these issues may be overcome with specialized strains and RNAi methods, working out these conditions is beyond the scope of this study.

5) Figure 2H provides a possible explanation to the observation that there is a lower presence of RNA Polymerase II in Hac-HSEs regulated genes (Figure 2G). Performing similar analyses using additional repression and activation epigenetic modifications would further establish this possibility.

Using publicly available ChIP-seq data for the “active” histone marks H3K4me3 and H3K36me3 and the “silent” histone mark H3K9me3, we compared these marks at Hac-HSE versus Hin-HSE genes. Consistent with our original observation of generally less Pol II and more repressive H3K27me3 at Hac-HSE genes, we find less of the active and more of the repressive marks at Hac-HSE genes compared to Hin-HSE genes and these results are presented as a new supplementary figure, Figure 2—figure supplement 1.

6) The authors should not use the word "domestication" to describe how Helitrons could be co-opted or acquired for other functions in the *C. elegans* genome. This word has specific meanings in the evolutionary genetics literature (e.g. a series of robust, controlled, evolutionary changes towards an often human-associated association) that are not supported here. The authors have beautiful data that do not need to use this loaded term.

The word “domestication” has been removed throughout.

7) I think that the Discussion is a bit long and parts of it are redundant with or can be included in the Introduction (if necessary). Please tighten both a bit.

We have revised the Discussion and Introduction as suggested.

8) The authors do a good job of explaining and defending their results in the Discussion and do not good too far in interpretations. One claim that could be softened (slightly) is that selection could be acting on the Hac-HSEs within the *Caenorhabditis* lineage. I agree that it could be, but it could also be drift. The differences in abundance across and within *Caenorhabditis* species suggests that it could be selected because different phylogenetic lineages have similar abundances (with unique complements of genes, probably) but it does not have to be selected in all cases. In *C. elegans*, the authors can look for signatures of selection in and around Hac-HSEs vs. Hin-HSEs (Tajima's D or the like) to look for support. Similar analyses can be done with C. briggsae using the ~40 wild isolates with sequence available. These data will support their claims of selection over drift.

We have modified the Discussion to further avoid the impression we are claiming selection and offer alternative possibilities, as suggested.

As advised, we have performed Tajima’s D calculations for Hac-HSE and Hin-HSE genes in *C. elegans* and *C. briggsae,* using available wild isolate sequence data. We calculated Tajima’s D over HSF-1 peak summits that overlap either Hac- or Hin-HSEs upstream of our defined Hac- and Hin-HSE genes and provide these scores in new columns in Supplementary files 3 and 4. From this, seven Hac-HSE genes (*col-162, str-30, ZK1053.1, ZK1053.2, srt-28, R02C2.6, oac-55*) and three Hin-HSE genes (*F57A8.1, Y38H6C.8, bath-36*) have scores in the bottom 5% of genome-wide Tajima’s D scores (calculated over 10-kb sliding windows in 100-bp steps), which could be interpreted as signatures of selection although we believe future work will be needed to make those conclusions.

We also determined the lowest Tajima’s D scores found in “promoter” regions 2.5 kb upstream of all Hac- and Hin-HSE genes in *C. elegans* and *C. briggsae*, as we do not have HSF-1 ChIP data for *C. briggsae* to center the analyses. For *C. elegans*, this results in 12 Hac-HSE genes and 22 Hin-HSE genes with scores in the bottom 5%. For *C. briggsae*, this results in two Hac-HSE genes and 30 Hin-HSE genes with scores in the bottom 5%. These scores have been added as new columns in what is now Supplementary file 8.

9) While I concur that this conclusion is true at face value, in that helitron insertions modify gene expression patterns when the animals are exposed to heat shock, it is less convincing to me that this has actually led biologically important changes in heat shock response in an adaptive way to benefit the fitness of the organisms. I certainly agree with the idea that helitron insertions (aka mutations) are likely to confer effects that manifest under heat shock; given what is known about the distribution of mutational effects in most organisms, nearly all such mutations will be deleterious or selectively neutral. The polymorphism observed among wild isolates could reflect helitron insertions that ended up being non-deleterious, or weakly deleterious but only under rare conditions of heat shock, thus allowing them to persist and not yet have been weeded out of the population by purifying selection. The fact that comparisons across species show unique effects reinforces the idea that lineage-specific rounds of helitron activity provides a mutational pressure that creates such novel elements with corresponding influence on neighboring genes, but that they usually are evolutionarily short-lived and may not confer any fitness benefit. The above "criticism" aside, the authors are generally quite cautious in not ascribing explicitly a fitness benefit to the helitron-mediated effects on flanking gene expression under heat shock, which I appreciate. Nonetheless, some of the language that the heat shock response is "rewired" leaves implicit the suggestion that this is an adaptive rewiring, which need not be the case. I think it would be valuable to make this point more explicit and more clearly articulate the mutational pressure of helitrons with respect to the potential distribution of fitness effects, with those persisting cases likely being neutral, weakly or conditionally-deleterious, or possibly beneficial; distinguishing among these possibilities would require additional study.

We appreciate these insights and the Discussion has been updated to include these points. Furthermore, the title has been changed to use the term “Diversification” instead of “Rewiring.”

10) It could also be interesting to relate (briefly) the polymorphism among wild isolates to the potential genetic basis to GxE interactions in phenotypes and transcriptome patterns. The enrichment of helitron insertions with specific GO categories is interesting, but could simply reflect sequence-biased mechanisms of insertion. For example, if GPCRs are enriched in certain sequence motifs that are conducive to helitron insertion, then the association of helitron with GPCR could simply be a by-product of the insertional bias. Alternatively, it could be that selection against helitron insertion is weaker for the large multigene family of GPCRs, and so it takes longer for them to be weeded out of the population to allow the helitron-GPCR association to reflect such less negative effects. These kinds of explanations for the GO enrichment analysis would be helpful so as not to mislead readers into thinking that the enrichment must reflect some adaptive story of selective retention in e.g. GPCRS.

To explore the possibility that the specific GO categories associated with HS-induced Hac-HSE genes might be due to insertional biases or insertions being better tolerated in large multigene families, we performed GO enrichment analyses for ALL Helitron-associated genes (both with and without HSEs), as well as genes that do not respond to HS and observed non-overlapping GO categories. While this seems to argue against sequence biased mechanisms of insertion as a predominant explanation for the specific GO categories associated with HS-induced Hac-HSE genes, we have revised the Discussion to consider this possibility.